# Flow patterns, hotspots and connectivity of land-derived substances at the sea surface of Curaçao in the Southern Caribbean

Vesna Bertoncelj[1,2], Furu Mienis[1], Paolo Stocchi[3,4], Erik van Sebille[2]

[1] Ocean Systems Department, NIOZ, Royal Netherlands Institute for Sea Research, the Netherlands
[2] Institute for Marine and Atmospheric Research Utrecht, Department of Physics, Utrecht University, Utrecht, The Netherlands
[3] Dipartimento di Scienze Pure e Applicate, Università degli Studi di Urbino "Carlo Bo", 61029 Urbino, Italy
[4] Institute for Climate Change Solutions, Via Sorchio, 61040 Frontone, Pesaro e Urbino, Italy

*Correspondence to*: Vesna Bertoncelj (vesna.bertoncelj@nioz.nl)

**Abstract.**

The South Caribbean Island of Curaçao is abundant in coral reef communities, but they are declining. Land-derived nutrients and pollutants are a potential contributing factor to this decline, since these substances after entering the ocean, can be transported towards reef sites by ocean currents. To study the movement of the substances and their potential impact on coral reefs, we developed SCARIBOS, a hydrodynamic model of the South CARIBbean Ocean System, with a 1/100° resolution, covering the period from April 2020 to March 2024 (excluding spin-up time) to analyse flow patterns within that period around the close proximity of Curaçao. SCARIBOS is used as hydrodynamic input for Lagrangian particle tracking analysis with the Parcels framework, where we assess the distribution of positively buoyant substances and explore connectivity within Curaçao's coastlines as well as with nearby regions of Aruba, Bonaire, the Venezuelan islands, and a portion of the Venezuelan mainland. Results reveal two dominant processes: the northwest-directed Caribbean Current and weaker cyclonic eddies moving in the opposite direction. These flow patterns influence hotspot locations, with higher accumulation of positively buoyant substances occurring during eddy events. Our analysis also highlights increased particle accumulation of land-derived substances in the northwest of Curaçao, corresponding to the prevailing currents. While the focus is on land-derived nutrients and pollutants, this methodology can be extended to study other particle types such as plastic debris and coral larvae, providing valuable insights for marine conservation efforts and environmental management.

# 1 Introduction

The island of Curaçao, located in the Southern Caribbean Sea (12°N latitude and 69°W longitude), features one of the Caribbean's most pristine and well-preserved shallow water fringing reef systems along its southern coastline. However, previous studies have reported a significant decline in coral cover, with more than 50% of living corals on shallow reefs lost between the 1970s and the early 2010s (e.g., Bak et al., 2005; Vermeij et al., 2011; Waitt Institute, 2017). These rates of decline vary across the island. Land-derived substances are known to be one of the causes of coral reef decline. Coral reefs, as sessile organisms, depend on environmental nutrients for survival and are susceptible to accumulating pollutants, bacteria, and viruses from their surroundings originating from various sources, such as urban areas, agricultural runoff, and nutrient-rich

groundwater (e.g., Dubinsky and Stambler, 1996; Fabricius, 2005; van Dam et al., 2011). For example, sewage systems with high inorganic nutrient content can lead to increased algae growth on corals and a higher likelihood of coral disease outbreaks

(e.g., Wear and Thurber, 2015). Areas with higher concentrations and longer residence times of these substances may cause increased stress on coral reef communities.

Coral reefs are not just impacted by local sources of pollution, but also by broader environmental changes and anthropogenic activities. Pollutants from distant sources can affect local coral reefs as ocean currents transport them over long distances. While several ocean circulation models simulate the Caribbean region's ocean dynamics, their spatial resolution is often

insufficient to capture the detailed dynamics affecting coral reefs locally. For instance, Lin et al. (2012) utilized a regional ocean circulation model with a 1/6° horizontal spatial resolution for the entire Caribbean Sea, while Sheng and Tang (2003) focused on the western Caribbean Sea, excluding Curaçao. Other models, such as those by Jouanno et al. (2008) with a 1/15° horizontal spatial resolution and van Der Boog et al. (2019) with a 1/12° horizontal spatial resolution also inadequately represent Curaçao, often as only one or two land grid cells. The highest-resolution model to date, the Regional US Navy

Coastal Ocean Model (NCOM), operated by the Fleet Numerical Meteorology and Oceanography Center (FNMOC) for the Americal Seas (product AmSeas) at a horizontal resolution of 1/30° (Barron et al., 2006), still fails to accurately represent Curaçao's coastline and completely neglects Klein Curaçao, limiting its usefulness in capturing critical dynamics around the island. These are just a few examples, but so far, no finer resolution model has been created to study the regional, intra-island ocean dynamics around Curaçao. These limitations highlight the need for finer resolution models to better understand regional

dynamics around Curaçao, understand the common pathways of land-derived substances and their hotspots, and provide a framework for similar studies in other islands or regions.

In our study, hotspots are defined as areas where substances are more likely to accumulate than in other areas, potentially leading to increased stress on coral reefs. To address the challenges outlined above, our research investigates the dominant surface ocean current patterns and substance transport pathways around Curaçao from April 2020 to March 2024, while also

considering the influence of monthly to inter-annual variability on these dynamics. Additionally, we determine the vertical extent of the surface current on a monthly average temporal scale to assess whether the surface layer accurately represents the overall surface dynamics. Specifically, we aim to answer the following research questions that are crucial for understanding how these hotspots form and affect coral reef communities: (1) How variable are the ocean current patterns around Curaçao from April 2020 to March 2024? (2) How do these ocean currents affect the movement and distribution of non-degradable,

positively buoyant substances at the ocean surface around the island, contributing to the formation of hotspots? (3) How are Curaçao's coastal zones connected to each other, as well as to coastal areas in nearby countries? By exploring these questions, we aim to provide a comprehensive understanding of the dynamic processes driving substance hotspots, which is essential for guiding targeted conservation efforts and enhancing local environmental management.

To study substance movement and identify hotspots, we developed a 1/100° resolution hydrodynamical regional model of the South CARIBbean Ocean System, named SCARIBOS, using the CROCO regional ocean model (Auclair et al., 2023). The model simulation covers the period from December 2019, and for analysis purposes, we focus on the model output from April 2020 to March 2024, after accounting for the necessary spin-up time. SCARIBOS addresses the first research question on surface ocean patters and serves as the hydrodynamic input for Lagrangian particle tracking, facilitated by the Parcels framework (Delandmeter and Van Sebille, 2019). We use this framework to simulate particle movement over this period to identify substance hotspots, capture their monthly and inter-annual variations, and reveal connectivity within the studied area. Our approach builds on successful applications of CROCO and similar models in tracking pollutants and studying marine dynamics (e.g., Vogt-Vincent et al., 2023).

## 2 Methods

### 2.1 Description of study site

The island of Curaçao is 61 km long and 14 km wide at its widest point (Pors and Nagelkerken, 1998). It is situated approximately 60 km north of Venezuela, between the neighbouring islands of Aruba and Bonaire (Fig. 1). The northern coastline, exposed to persistent easterly trade winds, experiences high wave action and features rough cliffs, making it minimally populated. In contrast, the southern coastline is more sheltered, hosting the capital city of Willemstad and the main tourist areas. This coastline features calm seas and popular beaches known for their coral reefs and residential communities. To the southeast lies Klein Curaçao, a small uninhabited island separated by a channel approximately 10 km wide and 800 m deep. The bathymetry around Curaçao is varied, with steep slopes just offshore from the southern coastline and more gentle slopes on the northern coastline, as depicted in Fig. 1.

Curaçao has a semi-arid climate and experiences two distinct seasons: a wet season from October to December and a dry season from January to September (Meteorological station of Curaçao, 2016). These seasons are primarily defined by variations in rainfall, wind, and temperature throughout the year. During the wet season, Curaçao receives most of its annual rainfall, with average monthly precipitation reaching up to 100 mm. In contrast, the dry season features significantly less rainfall, often below 24 mm per month. Prevailing winds over Curaçao are trade winds flowing from the east and east-northeast, with typical wind speeds ranging from 5 to 8 m s$^{-1}$. The tidal regime around Curaçao is primarily semi-diurnal, with a mean tidal range of 30 cm (Pors and Nagelkerken, 1998).

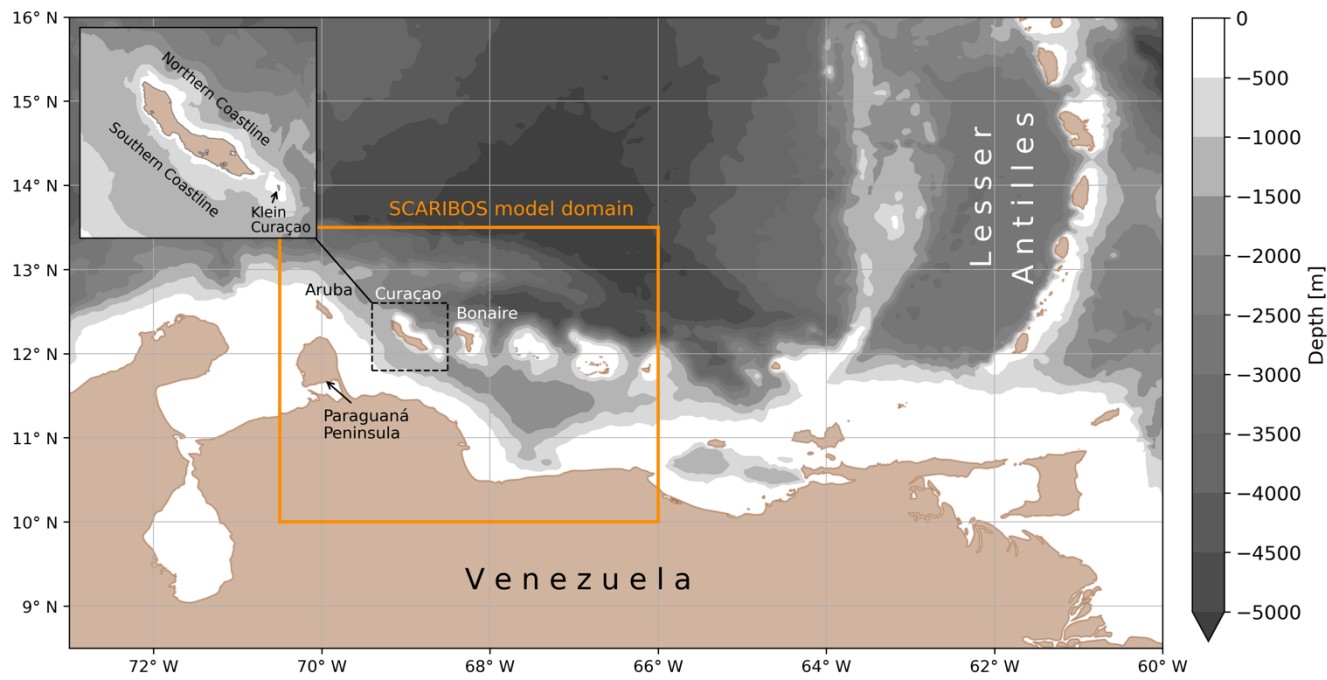

**Figure 1: Map of the Southern Caribbean Sea showing bathymetry from the global GEBCO 2023 dataset (GEBCO Compilation Group, 2023). The top-left inset zooms in on Curaçao, displaying detailed bathymetry derived from multibeam sonar collected during the RV *Pelagia* expedition 64PE500. The orange square indicates the domain of the SCARIBOS model. Major geographic features, including Curaçao, Aruba, Bonaire, and the Lesser Antilles, are labelled, along with the southern and northern coastlines**

**of Curaçao and Klein Curaçao.**

The ocean surface circulation in the Caribbean Sea is predominantly characterized by the westward flowing Caribbean Current, which is part of the upper branch of the Atlantic Meridional Overturning Circulation (Johns et al., 2002). This current is fed by two nearly equal sources of inflow water: north of 15°N from the returning Gulf Stream water and south of 15°N from tropical and South Atlantic origins (Richardson, 2005). South Atlantic water crosses the equator in the North Brazil Current

and flows northwest along the continental margin of South America, contributing to the Caribbean Current. The Caribbean Current exhibits spatial variability on seasonal and inter-annual scales, influenced by seasonal changes in the easterly trade winds (Chadee and Clarke, 2014; Chang and Oey, 2013; Wang, 2007) and fluctuations in freshwater input from the Amazon and Orinoco River plumes (Beier et al., 2017). Additionally, the El Niño Southern Oscillation impacts the Caribbean Current on an inter-annual scale (Beier et al., 2017).

As the Caribbean Current flows across the Caribbean Sea, it meanders and generates mesoscale eddies. These eddies exchange the mass and momentum of the Caribbean Current, thus potentially playing an important role in the North Atlantic circulation system (Carton and Chao, 1999). The eddies are predominantly anticyclonic in the centre of the Caribbean Sea and often

cyclonic at the boundaries (Richardson, 2005). Cyclonic and anticyclonic eddies are regular features in the Caribbean Sea, appearing at intervals of approximately every three months west of the southern Lesser Antilles (Carton and Chao, 1999).

They progress westward with an average speed of 0.12 m s$^{-1}$ (Carton and Chao, 1999) to 0.15 m s$^{-1}$ (Murphy et al., 1999), taking about 10 months to travel from the Lesser Antilles to the Yucatan Channel. They grow in amplitude as they move, and their vertical extent is often confined to the thermocline (Carton and Chao, 1999), but recent observations found some anticyclonic eddies to be confined to the isothermal layer too (Van der Boog et al., 2019).

The interplay of eddies and the Caribbean Current significantly influences the movement and distribution of substances around

115 the Caribbean Sea. On smaller scales, such as around the island of Curaçao, eddies can interact with the boundaries (e.g., coastline and islands), mixing the nearshore waters with the offshore (Richardson, 2005). These smaller-scale processes are critical to understanding local flow circulation and require fine resolution models in combination with detailed bathymetry to capture their complexity.

## 2.2 Hydrodynamic modelling of the Southern Caribbean Sea

We ran the multi-year 3D hydrodynamic model SCARIBOS with Coastal and Regional Ocean Community Model (CROCO version 1.3.1; Auclair et al., 2023). The model domain extends 492.7 km in zonal and 389.7 km in meridional directions, covering longitudes from 70.5°W to 66.0°W and latitudes from 10.0°N to 13.5°N, as indicated in Fig. 1. The horizontal resolution of the model is 1/100°, translating to a spatial resolution of approximately 1.1 km in both the meridional and zonal directions within the model's domain. The model is terrain-following over the vertical with 50 vertical sigma layers. The

vertical resolution is finer at the surface, with the thickness of the upper layer ranging from a few centimetres to approximately 4 meters. The model is run for 52 months from 1 December 2019 to 31 March 2024. Due to spin-up time the first four months (December 2019 to March 2020) are discarded from the analysis. A spin-up duration of four months was selected based on the assumption that the initial and boundary conditions (provided by the E.U. Copernicus Marine Service Information product, described below) allow the system to reach a quasi-equilibrium state. This duration allowed that the eddy kinetic energy (EKE)

stabilized across the model domain (Fig. S1), ensuring the system reached a steady state before the analysis period began. Model outputs include hourly averages of horizontal and vertical velocities, temperature and salinity, stored for every grid cell in the domain (upon request available at Bertoncelj, 2025a), and the portion of the hourly output for surface currents (u, v of surface layer only) and water level time series is available for download at Bertoncelj (2025a).

The bathymetry input for SCARIBOS is created through the integration of global product GEBCO with a spatial resolution of

135 15 arc-seconds (version 2023; GEBCO Compilation Group, 2023) and local bathymetry around Curaçao obtained using multibeam sonar during RV *Pelagia* expedition 64PE500 with a spatial resolution of 10 m, using linear interpolation. Smoothing of the bathymetry was performed using the CROCO TOOLS product (V1.3.1) to mitigate steep slopes that could cause instabilities in the model. Additionally, for more realistic flow around Curaçao, some adjustments are made to the land

grids specifically for Curaçao, Klein Curaçao and the neighbouring island of Bonaire. These adjustments are necessary to correct inconsistencies between the bathymetry-derived land-sea mask and the true coastline, ensuring more accurate representation of coastal features that significantly impact the formation and propagation of eddies around the islands. The adjustments to the land grid cells were manually performed based on satellite imagery.

Oceanographic initial and boundary conditions for SCARIBOS are generated using the E.U. Copernicus Marine Service Information product reanalysis GLORYS12V1 (Lellouche et al., 2021) with daily temporal resolution of variables salinity, temperature, horizontal current velocities and sea-surface height at spatial resolution of 1/12° and 50 Z-depth levels, which are interpolated into the CROCO grid with the use of CROCO TOOLS product (V1.3.1). The model uses a 'hot start,' initializing with horizontal velocities, salinity and temperature, interpolated from the GLORYS12V1 data and included in the initial state. Tidal forcing is derived from TPXO7-atlas (Egbert and Erofeeva, 2002). Atmospheric forcing is derived from ERA-5 global atmosphere reanalysis (Hersbach et al., 2020) with 1/4° horizontal resolution and hourly temporal resolution. Four major rivers are included in the domain, all discharging into the Southern Caribbean Sea from continental Venezuela: Tocuyo, Yaracuy, Tuy and Grande. Their discharge has a strong seasonal variability, with range of 2-6 $m^3$ $s^{-1}$ in Venezuelan dry months (FMA) and 10-18 $m^3$ $s^{-1}$ in Venezuelan wet months (MJJAS), based on a climatological river discharge dataset by Dai and Trenberth (2002). Stokes Drift is not included in the model.

Key numerics in SCARIBOS include the use of a nonlinear equation of state (Jackett and McDougall, 1995) which represents seawater density as a function of temperature, salinity, and pressure. For advection, a third-order upstream biased advection scheme is applied to lateral momentum, while a fourth-order compact advection scheme is used for vertical momentum. Lateral tracer advection employs a split and rotated third-order upstream-biased scheme, and vertical tracer advection uses a fourth-order compact advection scheme. The model uses a baroclinic time step of 50 s with 60 barotropic time steps for each baroclinic step. Surface turbulent fluxes are parameterized using the COARE3p0 bulk formulation (Fairall et al., 2003), incorporating gustiness effects (Godgrey and Beljaars, 1991) for improved representation of light winds and longwave radiation feedback from model sea surface temperature, while current feedback is disabled. Vertical mixing is parameterized using the K-profile parameterization (KPP) (Large et al., 1994). Radiative open boundary conditions are applied to baroclinic velocities and tracers (salinity and temperature) at the lateral boundaries, while characteristic methods are used for barotropic velocities to properly handle tidal forcing. Moreover, sponge layer is implemented at the lateral open boundaries to enhance viscosity and diffusivity, ensuring a smooth transition between boundary data and the model interior. Bottom friction is parameterized using quadratic friction with a log-layer drag coefficient ($C_d$ ranging between 0.0001 and 0.1).

## 2.3 Assessment of hydrodynamic model

The assessment of SCARIBOS is conducted for surface currents, salinity, temperature and water levels, focusing specifically on surface properties. This choice aligns with the scope of the study, which only uses the surface layer for particle tracking,

although the flow in deeper layers can be used for future analysis too. Three main assessment techniques are employed: (1) a comparison of surface currents simulated by SCARIBOS for the year 2022 with monthly total surface currents derived from the E.U. Copernicus Marine Service Information product GlobCurrent (Rio et al., 2014), which combines satellite geostrophic surface currents and modelled Ekman currents at a spatial resolution of 1/4°, (2) a comparison of water levels with available observations from a water level bubbler, and (3) a comparison of surface currents with a Teledyne RDI 600kHz Acoustic

Doppler Current Profiler (ADCP) data collected during the RV *Pelagia* expedition 64PE529. The letter two assessments are performed for January 2024, as ADCP data are only available for this period.

Additionally, the sea-surface temperature (SST) and salinity (SSS) fields simulated by SCARIBOS are assessed for the year 2022 using multi-observation datasets (referred to as MultiObs) from the E.U. Copernicus Marine Service Information, which combine in situ observations and satellite measurements (for SST: Guinehut et al., 2012, and for SSS: Droghei et al., 2016).

Furthermore, surface currents, SST and SSS are compared with the GLORYS12V1 model, which provides the initial and boundary conditions for SCARIBOS. These additional analyses are provided in the Supplementary Material (Figs. S2-S6).

Figure 2 compares monthly surface currents from SCARIBOS with monthly total (geostrophic + Ekman) surface currents from GlobCurrent for 2022. Here, SCARIBOS is interpolated to a 1/4° grid to match the grid of GlobCurrent. Across all months, the general direction of the surface current is comparable, with the flow across the entire domain predominantly moving west

to northwest. SCARIBOS effectively captures months with stronger currents, particularly in February, March, April, July and December 2022, where differences arise from the positioning of the core of the current. Notably, SCARIBOS simulates strong flow both south and north of the islands, while GlobCurrent shows stronger currents primarily north of the islands. These differences may stem from SCARIBOS resolving finer-scale flow interactions around the islands, including ageostrophic effects, whereas GlobCurrent, while incorporating the large-scale influence of islands through sea surface height observations,

does not resolve their small-scale effects explicitly. When comparing SCARIBOS with GLORYS12V1 (Fig. S2), a much stronger agreement is observed, with both models showing strong currents both north and south of the island. This is expected, as GLORYS12V1 provides the boundary conditions for SCARIBOS, indicating that the differences with GlobCurrent likely arise from the choice of boundary conditions.

During low-energy months, such as September and October 2022, both SCARIBOS and GlobCurrent show reduced energy

levels, with currents around Curaçao flowing in the opposite direction to the typical northwest flow. This is especially evident in September 2022, when GlobCurrent clearly displays a cyclonic eddy in the upper-central region of the domain. A similar cyclonic eddy is also present in GLORYS12V1 (Fig. S2).

Furthermore, the observed northwestward direction of the current aligns well with previous studies, supporting the pattern observed in our simulations. There is strong agreement between the surface current vectors simulated by SCARIBOS and those

estimated from Lagrangian surface drifters in the Caribbean Sea between 1989 and 2003 (Richardson, 2005). Jouanno et al.

(2008) reported a northwest-directed flow at 30 m water depth over six years, with stronger currents at the western boundary and near the Paraguaná Peninsula. A regional ocean circulation model by Lin et al. (2012) found similar northwestward currents at 2.5 m water depth, with reduced flow near the southeastern Venezuelan coast. Xu et al. (2024) identified a consistent northwestward current from 2010 to 2021, which aligns with our simulation results.

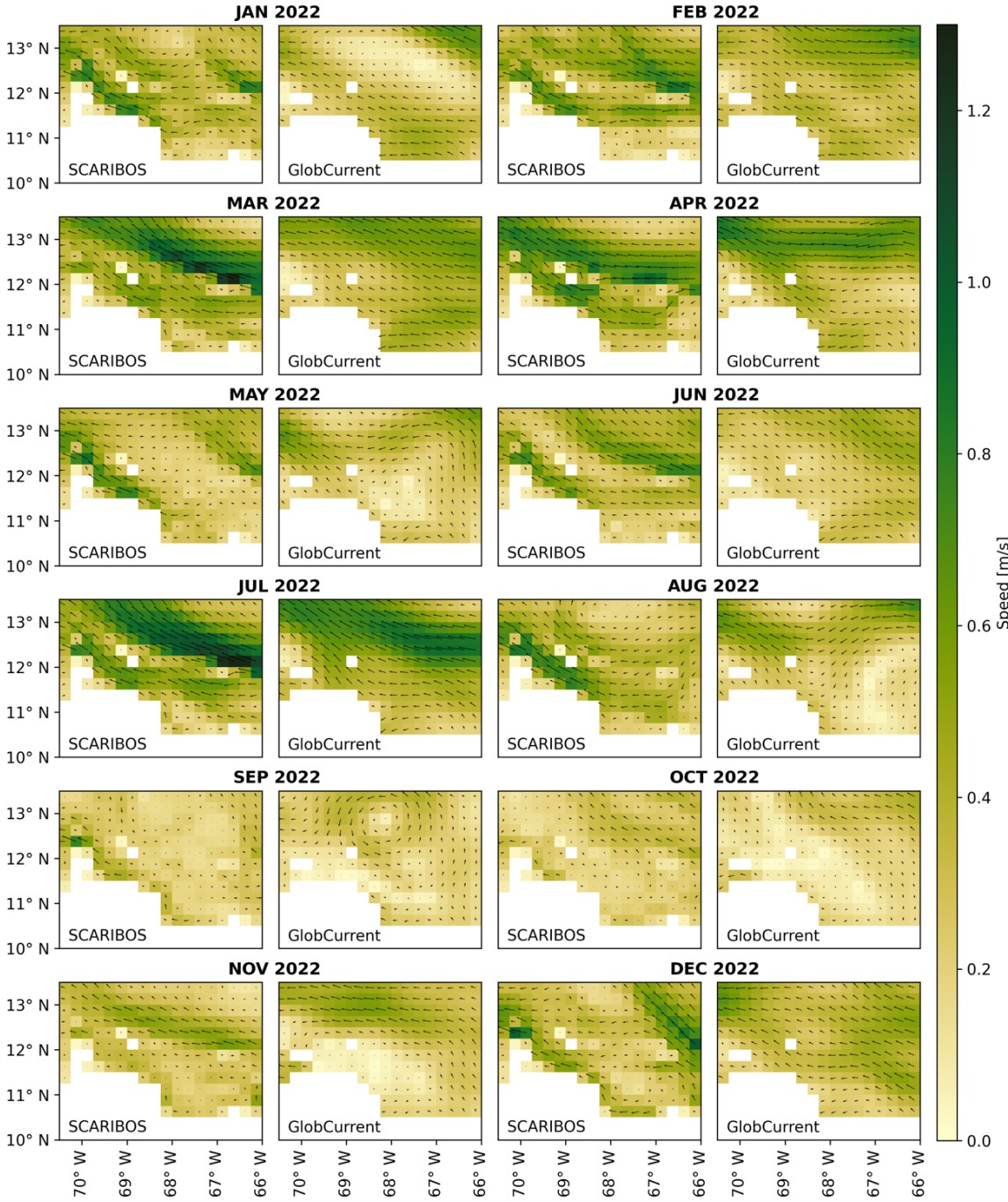

**Figure 2: Comparison of surface currents across the entire domain between SCARIBOS model output and Copernicus GlobCurrent Surface (GlobCurrent) for the year 2022. SCARIBOS surface currents are interpolated to a 1/4° grid to match the spatial resolution**

**of the GlobCurrent dataset. The colour scheme represents surface speed, with arrows indicating both direction (arrow shape) and**
**magnitude (arrow size). All plots share the same colour bar and arrow scale.**

The surface temperature and salinity fields from SCARIBOS show good agreement with the multi-observation datasets, particularly for sea surface temperature (SST), while larger differences are observed for sea surface salinity (SSS). To enable a direct comparison, SCARIBOS was interpolated to the respective spatial resolutions of the observational and reanalysis products (1/8° for MultiObs for both SSS and SST, and 1/12° for GLORYS12V1).

The SST comparison with MultiObs (Fig. S3) shows overall consistency, with differences primarily arising along the Venezuelan coastline, particularly in April 2022, when SCARIBOS simulates cooler coastal SSTs. Notably, around the Paraguaná Peninsula, SCARIBOS captures finer-scale coastal features, including the influence of the large bay in the region, which is not well represented in MultiObs due to its coarser resolution. The comparison of SST with GLORYS12V1 (Fig. S4) further confirms this agreement, revealing similarly detailed SST structures in both SCARIBOS and GLORYS12V1. However,
the Venezuelan coastline remains a region of discrepancy, with SCARIBOS having cooler coastal surface temperatures. Despite these localized differences, the overall SST agreement between SCARIBOS and both datasets is strong.

For SSS, larger differences are observed when comparing SCARIBOS with MultiObs (Fig. S5), particularly around the Paraguaná Peninsula. This is expected, as MultiObs does not represent all coastal features such as the large bay in this region due to its coarse resolution. Additionally, differences appear in the northern half of the domain, where SCARIBOS generally
simulates fresher surface waters, especially in months July to November 2022. However, when comparing SCARIBOS to GLORYS12V1 (Fig. S6), these discrepancies are minimal, with differences primarily confined to the Paraguaná Peninsula. This suggests that the observed differences from MultiObs are largely due to boundary conditions, as SCARIBOS shows strong agreement with GLORYS12V1, the provider of its initial and boundary conditions.

The water level assessment is performed by comparing model outputs with observed tidal data from a station at Bullenbaai
(12.187° N, 69.019° W). Hourly water levels from the SCARIBOS model at the nearest grid point are plotted against the observed tidal data for the entire month of January 2024. Additionally, contributions of 10 main tidal constituents (M2, S2, N2, K1, O1, P1, Q1, K2, M4, M6) are computed and compared between the observations and the SCARIBOS output.

Figure 3 presents the comparison of water levels for the entire month of January (3a), together with tidal amplitude (3b) and phase (3c) for the 10 main tidal constituents. Overall, the results demonstrate a strong agreement between the observations and
the SCARIBOS model output, with the model accurately capturing tidal dynamics, including the neap and spring tides, and reproducing their amplitude well.

The comparison of tidal constituents also indicates that the model captures the dominant contribution of amplitude from the K1 constituent (lunar diurnal tides) with centimetre-level precision. While the model slightly overpredicts the amplitude of the first four constituents (M2, S2, N2, K1) and K2, it underpredicts the rest. The phase predictions for most constituents are also well aligned with observations, suggesting that SCARIBOS effectively models tidal propagation across the region towards Curaçao.

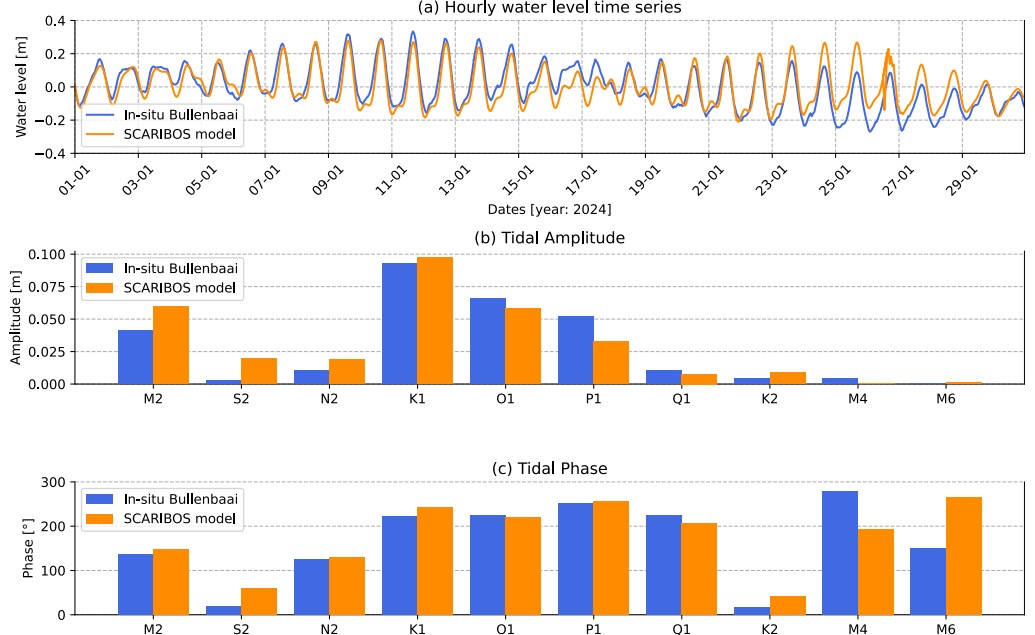

**Figure 3: Comparison of water levels at Bullenbaai (12.187° N, 69.019° W) between SCARIBOS model output (in orange) and the observations from a bubbler sensor (in blue) for the entire month of January 2024. (a) Hourly comparison of water levels over the month. (b) Comparison of tidal amplitude for the 10 main tidal constituents (M2, S2, N2, K1, O1, P1, Q1, K2, M4, M6). (c) Comparison of tidal phase for these constituents.**

The assessment of surface currents is performed using data collected in the period between 4 and 22 January 2024 measured with a vessel-mounted ADCP. Measurements used for this assessment were collected when the ship was in motion along transects from the coast to up to 12Nm from the island of Curacao. Post-processing on the ADCP measurements is applied to correct for ship navigation and heading changes, and poor-quality data are discarded. For comparison with the surface currents of the SCARIBOS model, the ADCP data from the uppermost layer with a bin centre depth of 20.22 m are selected. The SCARIBOS model output was averaged over the same period (4 to 22 January 2024) to align with the ADCP sampling timeframe. The comparison was conducted visually, focusing on both the direction and speed of surface currents between the

ADCP data and the averaged model output. For this purpose, SCARIBOS velocity quivers were interpolated to the locations
of the ADCP data to allow for direct comparison.

Figure 4 reveals that the surface current direction in the SCARIBOS model (4a) aligns well with observations (4b), particularly around the southern coastline of the island where currents are predominantly in the northwest direction. It is important to note that the observations, which are snapshots from a three-week period during January 2024, are compared against the SCARIBOS model's average over this period. This difference in temporal resolution can explain some minor deviations in
current direction. Additionally, the SCARIBOS model represents currents in a surface layer just a few meters deep, whereas the ADCP data represent currents at a depth of 20.22 meters. Despite this discrepancy, both datasets show strong agreement in the current magnitudes. The currents also show similar patterns in their strength, as in both cases they are stronger further southwest from the southern coastline.

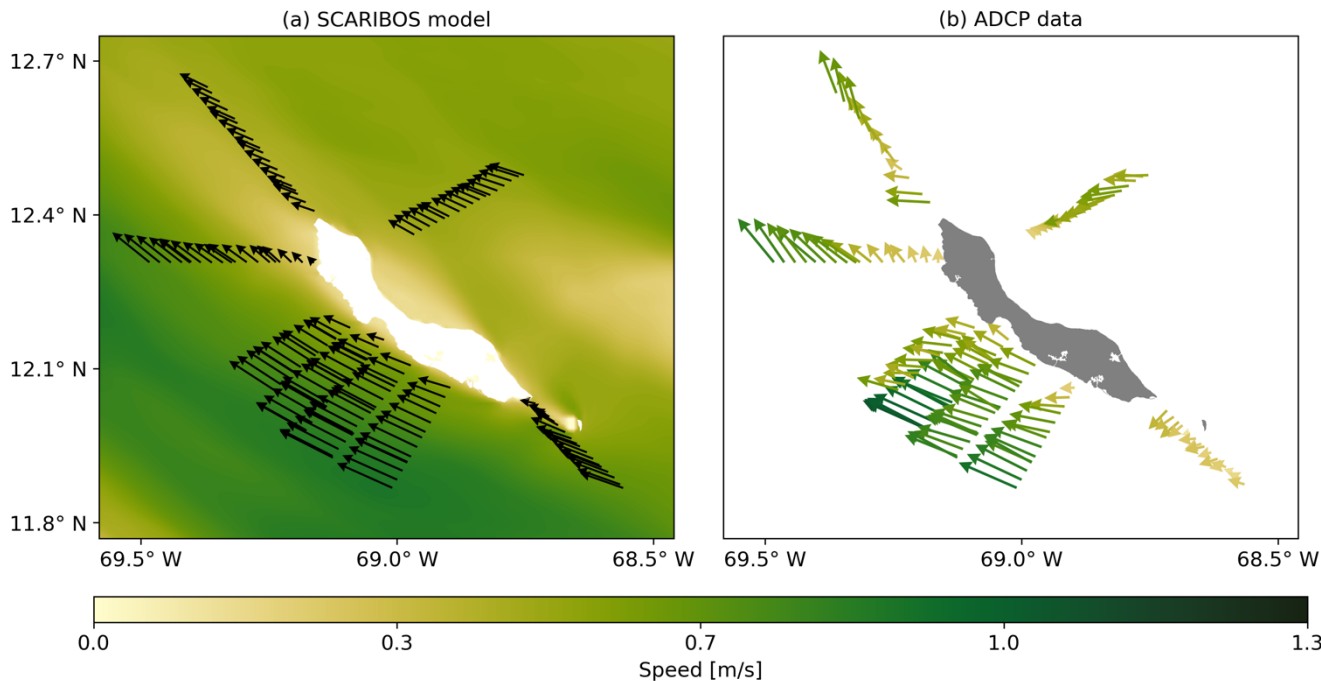

**Figure 4: Comparison of surface currents in the confined area around Curaçao between (a) SCARIBOS model output and (b) observations from a vessel-mounted Acoustic Doppler Current Profiler (ADCP) installed on RV *Pelagia* during the 64PE529 expedition. Both datasets are derived from the period between 4 and 22 January 2024. The colour scheme represents surface speed, with arrows indicating both direction (arrow shape) and magnitude (arrow colour in case of observation data). Both plots share the**
**same colour bar and arrow scale.**

Both datasets reveal the shadowing effect of the island, where currents are weaker in the northwest of the southern coastline and converge at the island's tip. However, a notable deviation is observed southeast of the island, between Curaçao and Klein Curaçao, and south of Klein Curaçao, where the observed current speeds are significantly lower (by half) and the direction deviates towards the southwest, while the model shows northwest flow. This deviation is not unexpected, as small-scale variations and local anomalies may not be fully captured in this analysis.

These results confirm that SCARIBOS accurately simulates surface-level dynamics, making it a reliable tool for tracking surface currents and tides around Curaçao. The agreement in tidal amplitude and phase, as well as the consistency between modelled and observed surface current patterns strengthens confidence in the model's ability to reproduce key hydrodynamic processes needed for accurately simulating particle movement and consequently assessing how surface water dynamics moves the substances around coral reef communities on Curaçao. Some differences in surface currents, SST and SSS compared to the MultiObs products, which combine observations and satellite measurements, are present. Since SCARIBOS shows strong agreement with its boundary condition and initial condition provider, GLORYS12V1, this suggests that substantial part of these differences originates from the GLORYS12V1 model rather than the SCARIBOS model itself.

### 2.4 Lagrangian particle tracking

Particle tracking simulations using the Parcels v3.0.3 framework (Delandmeter and Van Sebille, 2019) are conducted to model the movement of passive particles, representing nutrients and pollutants, using hourly velocity fields from the uppermost layer of the SCARIBOS model. These velocity fields do not include Stokes Drift. The particles, simulating non-degradable, positively buoyant substances, move with the surface flow conditions. Three distinct scenarios are investigated: Hotspots around Curaçao, Intra-island connectivity, and Coastal connectivity, each varying in particle release locations, tracking duration, and analysis methods. Similar approaches using Parcels have been employed in numerous studies, such as modelling marine debris sources to remote islands (Vogt-Vincent et al., 2023), simulating coral larvae connectivity using surface currents (Vogt-Vincent et al., 2024), assessing larvae dispersal in French Polynesia (Raapoto et al., 2024), and studying inter-island connectivity in the Hawaiian Islands and Guam (Carlson et al., 2024; Hirsh et al., 2023).

In Scenario 1 (*Hotspots around Curaçao*), particles are released within a square area surrounding Curaçao to identify potential hotspots. The area extends 1/2° in each direction from the centre of the island. 9673 particles are released every 12 hours with a spatial resolution of 1/100° in both longitude and latitude, excluding the land grid cells, depicted in Fig. 5a. To optimise computational efficiency, each particle is removed from the simulation 30 days after the release. The analysis involves calculating the normalized unique particle count to identify hotspots. The area is gridded with the same extent as the particle release area, divided into 100 by 100 grid cells. For each particle trajectory, a count is recorded when a particle enters a grid cell at least once. These counts of unique particle visitations are then summed for each grid cell and normalized by the total number of particles. This analysis is conducted separately for each month, providing a spatial representation of unique particle

visitation and highlighting areas with a higher likelihood of particle accumulation. Additionally, a time-averaged map is created by aggregating data from all months to overview hotpots over the entire simulation period. Monthly and time-averaged maps are then plotted on a logarithmic scale to visualise a wide range of values and identify both high-density and low-density areas within a studied domain.

In Scenario 2 (*Intra-island connectivity*), particles are released in a ribbon two coastal grid cells away from the islands of Curaçao and Klein Curaçao to analyse the connectivity around the island itself. Particles are released every 12 hours at 8 coastal zones, defined based on the 2017 report on the state of Curaçao's coral reefs (Waitt Institute, 2017). The zones are depicted in Fig. 5b along with their respective number of particles released per zone at each release interval. During the post-processing stage, particles are tagged when they reach another zone along the Curaçao's coastline. The destination areas of each zone are depicted in Fig. 5b. The analysis focuses on determining the connectivity within Curaçao by generating a histogram of percentage of particles reaching each zone within a specific time window.

In Scenario 3 (*Coastal connectivity*), particles are released at every coastal grid cell of the domain, excluding the coast of Curaçao, with a spatial resolution of 1/100°, in order to analyse the connectivity between other coastlines in the model domain and Curaçao. The regions considered are Aruba, Bonaire, Venezuelan islands, and the Venezuelan mainland, depicted in Fig. 5c. These particles represent land-based substances originating from afar that might reach Curaçao's coastal reef communities. Particles are released in a ribbon two coastal grid cells away from the land cells every 12 hours. In the post-processing stage, particles are tagged when they reach the proximity of Curaçao's coastline (the purple area around Curaçao highlighted as destination area in Fig. 5c). As a result, we show a histogram depicting the percentage of particles reaching Curaçao each month from each region.

In all scenarios, particles are released at regular intervals throughout each month, starting from the first day until the last day of the month. The internal particle simulation timestep is set to 5 min and trajectories are archived every hour. In Scenarios 1 and 2, each particle is tracked for a duration of 30 days from its release date, meaning that particles released at the end of a month continue to be tracked into the following month. The analysis focuses on particles released within each calendar month, even though their trajectories may extend beyond the initial month of release. In Scenario 3, particles are released every 12 hours for the duration of a given month and tracked until they leave the domain or until the end of the available SCARIBOS model output (until March 2024). This approach ensures that particles far from Curaçao can also reach Curaçao, even over longer timescales. Most particles exit the domain within 30 days of release. Additionally, the simulation may be terminated early if the number of particles remaining in the domain falls below 1%, due to computational demands, ensuring that the analysis focuses on significant particle dynamics.

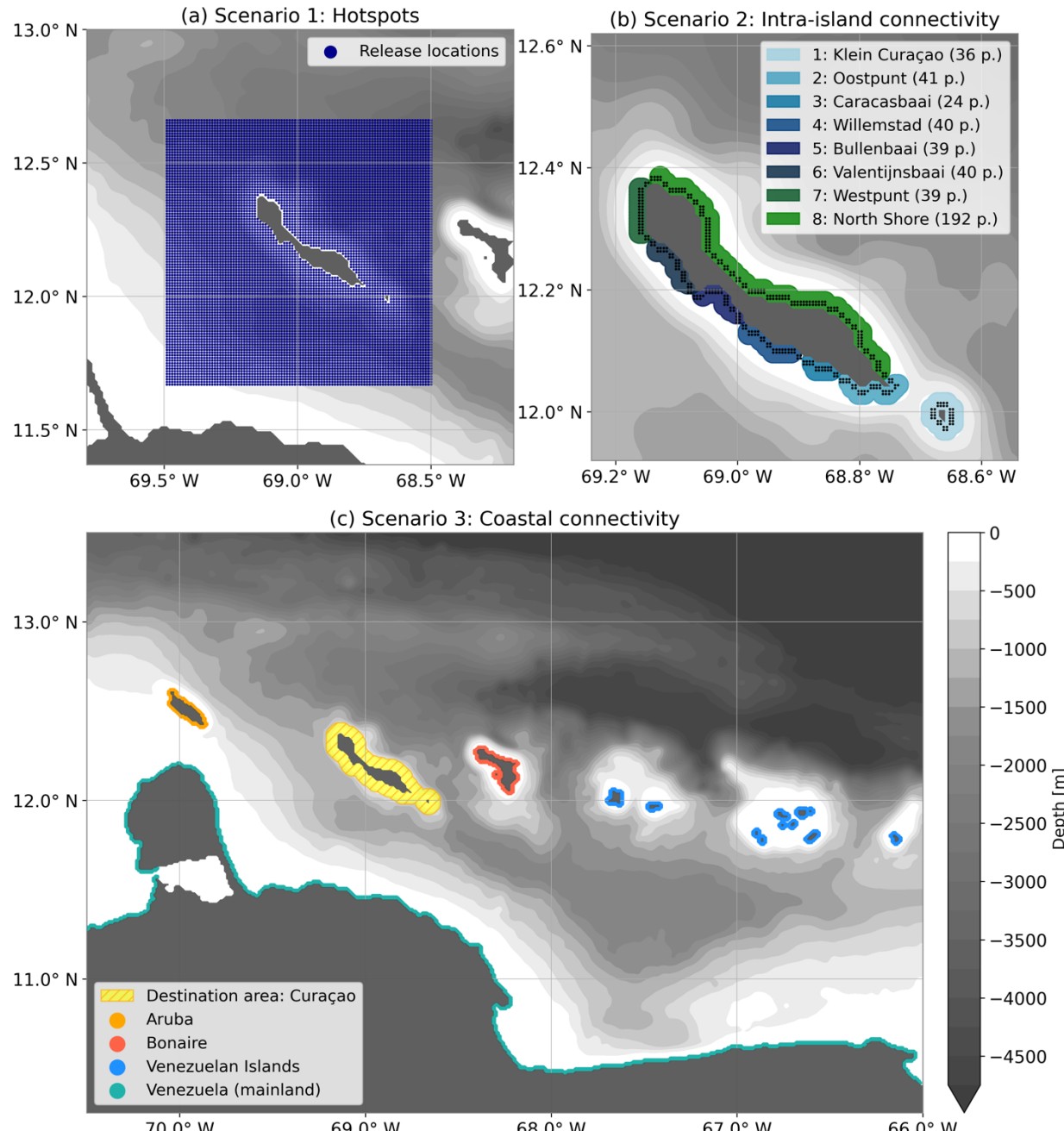

**Figure 5: (a) Particle release locations for Scenario 1, identifying potential hotspots. (b) Particle release locations (dots) for Scenario 2, colour-coded for by coastal zone, with surrounding areas indicating potential destinations for connectivity analysis. (c) Particle release locations (dots) for Scenario 3, colour-coded by coastlines of each source area. The destination area highlighted around Curaçao represents the region within which particles are tagged as reaching Curaçao coast.**

## 3 Results

### 3.1 Ocean currents around Curaçao

The average surface currents over the entire simulation period from April 2020 to March 2024, simulated using the CROCO model, indicate predominant west to northwestward currents across the model domain (Fig. 6a). Furthermore, the presence of islands significantly influences the flow patterns, creating areas of reduced current strength in their lee. This shadowing effect is most pronounced in the wake of Aruba (70°W, 12.5°N) and is also clearly observed in the wake of other islands within the domain.

The monthly average of surface currents around Curaçao is depicted in Fig. 6b, revealing seasonal to inter-annual variability in surface current speed and direction. The surface flow from December to March is predominantly unidirectional and northwestward, and it is often the strongest of the year. This strong unidirectional northwestward current also occurs in at least one of the months between July and September each year. Starting from April or May each year, deviations from the average current occur, resulting from cyclonic eddies that pinch off the main Caribbean surface Current. This creates an opposing southeastward current along the southern coastline of Curaçao, which is also weaker in strength. The presence of these cyclonic eddies can persist for up to several months. In 2023, the presence of these eddies is particularly prolonged, lasting almost continuously for up to eight months.

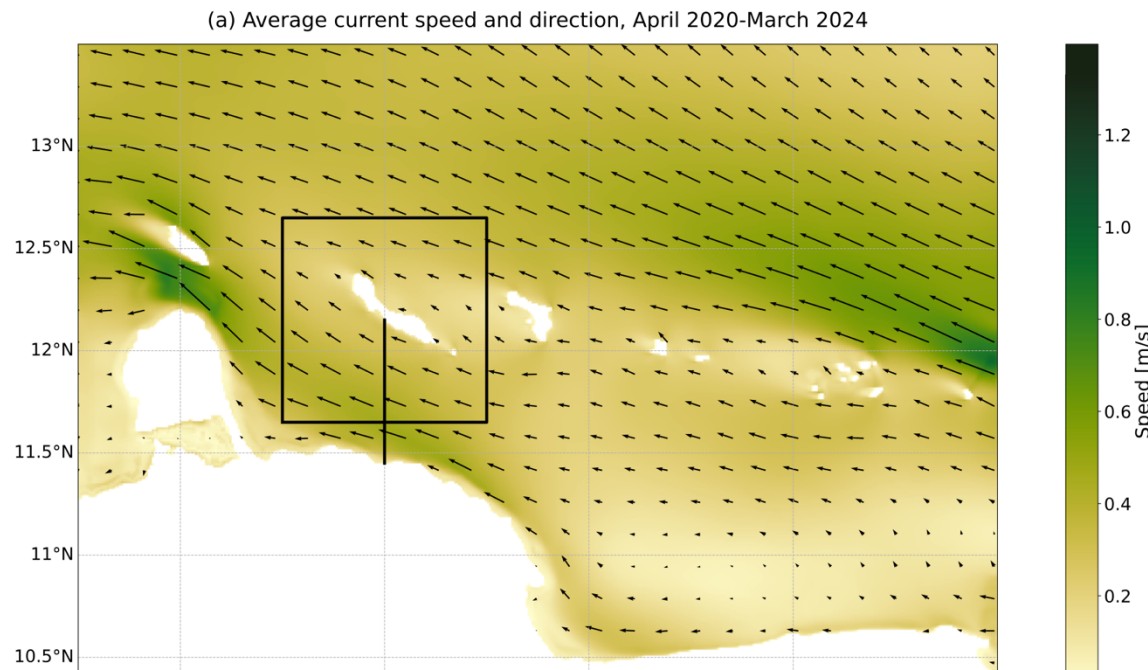

(a) Average current speed and direction, April 2020-March 2024

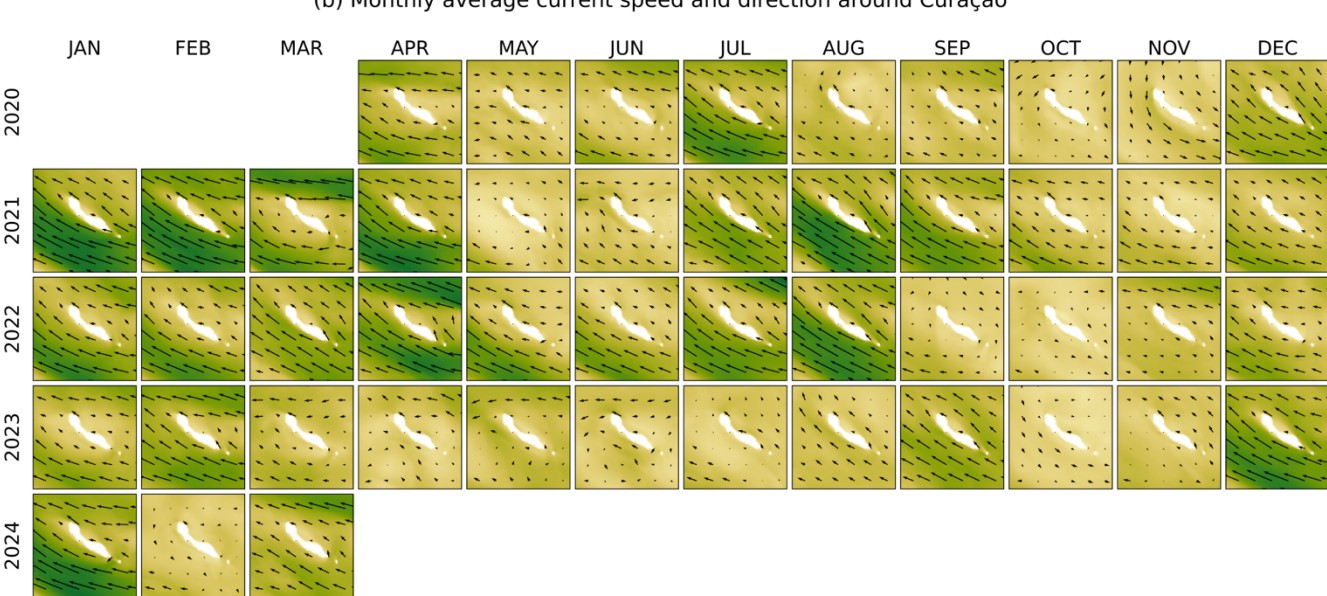

(b) Monthly average current speed and direction around Curaçao

**Figure 6: (a) Average surface current speed and direction of the entire domain for the entire simulated period from April 2020 until**
**March 2024. Square indicates the area zoomed for further analysis on monthly variations and the line indicates the meridional cross-**

**section, used for analysis of the currents along the meridional cross-section at 69°W longitude between Curaçao and continental Venezuela. (b) Average surface current speed and direction for each month of the simulated period. Colour scale is the same as top.**

We conducted an analysis of the currents along the meridional cross-section at 69°W between Curaçao and continental Venezuela, focusing on the zonal currents for the simulation period from April 2020 to March 2024 to understand the depth extent of the surface current and evaluate whether the surface layer provides an adequate approximation of the general surface dynamics, considering our model employs sigma-layer vertical discretization.

The average zonal velocity depth profile (Fig. 7a) reveals a westward-directed surface current that extends to both the Venezuelan coast and Curaçao's shore, with depths reaching 50 m water depth near the shores and up to 400 m water depth in the centre of the channel. An eastward counter current is present at deeper depths, which is most pronounced along the continental shelf of Venezuela and weakest along the Curaçao's steep slope. Additionally, a weak westward current is present at depths below 1200 m.

Monthly variations of the zonal velocity along the meridional cross-section (Fig. 7b) reveal a dynamic two- to three-layer system, with a distinct westward surface current often overlaying a deeper counter current, as shown in the average profile (Fig. 7a). However, this system sometimes undergoes significant changes under specific scenarios. Firstly, during periods when the northwestward directed surface current dominates, it can extend to greater depths than average, leading to the absence of a distinct undercurrent. An example of this is in January 2021, when the current was unidirectional and strongest at the surface, yet it extended all the way to the seafloor. Secondly, during months with a presence of a cyclonic eddy (Fig. 6b, October-November 2020 and April-July 2023), the surface current weakens and the entire water column exhibits a barotropic easterly flow layer.

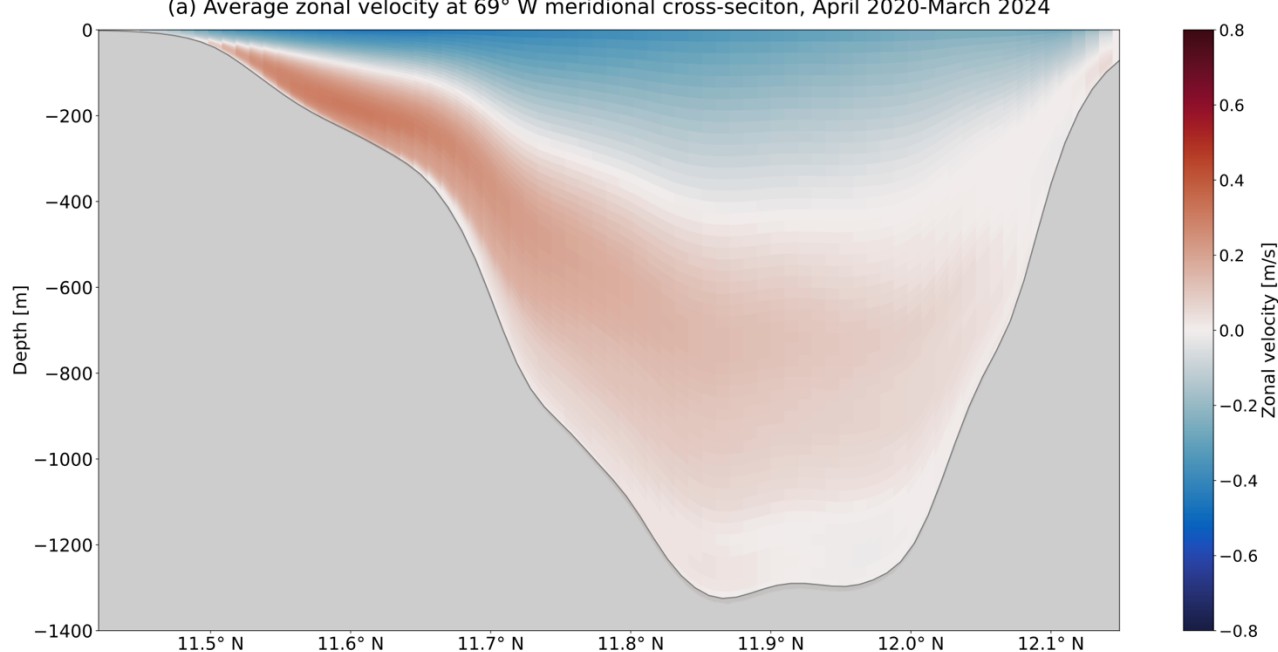

(a) Average zonal velocity at 69° W meridional cross-seciton, April 2020-March 2024

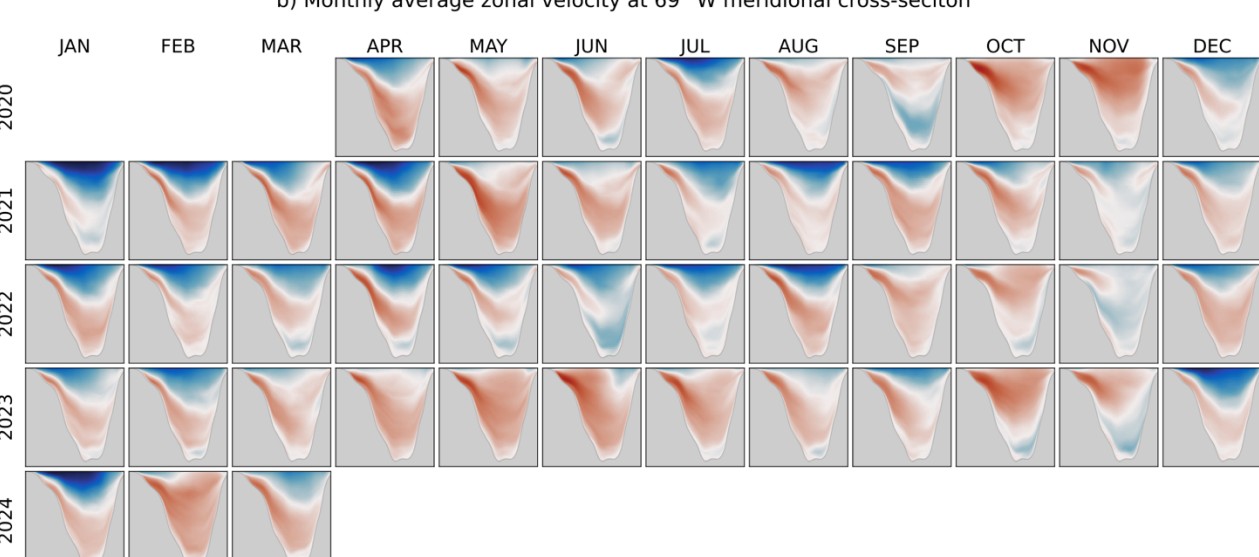

b) Monthly average zonal velocity at 69° W meridional cross-seciton

**Figure 7: (a) Average zonal velocity of the entire simulation period from January 2020 until December 2023 along the meridional cross-section at 69°W longitude between Curaçao and continental Venezuela. (b) Average zonal velocity along the meridional cross-section for each month of the simulated period. Colour scale is the same as top.**

Our primary objective is to determine the depth extent of the surface current on a monthly average temporal scale to assess the representation of overall surface dynamics by the surface layer. By identifying the depth range with minimal variations, typically presenting as a two- to three-layer system, we show that the top layer of our sigma-layer model effectively approximates general surface currents, corresponding to the first layer in this system. This assessment supports our use of the surface layer in the SCARIBOS model for further investigations into the movement of substances. The top layer of the model simulation provides a reasonable approximation of average conditions in the upper 10-20 m of water depth, which contributes to our understanding of the dynamics potentially affecting shallow water coral reefs.

## 3.2 Hotspots around Curaçao

Hotspots are areas where particles, originating either from the land or the open ocean, are more likely to accumulate relative to other regions around Curaçao. The time-averaged analysis of normalized unique particle counts (Fig. 8a) reveals a distinct pattern along the island's coastline. The northern coastline, particularly the northwestern part of it, shows high particle counts, indicating that particles from various origins frequently visit this area. On the contrary, the southern coastline exhibits a pronounced shadowing effect, where low particle count is present along the entire southern coastline. This suggests that the southern coastline is largely disconnected from offshore particle influence, as offshore particles rarely reach this region.

## a) Normalized unique particle count, time average over entire period

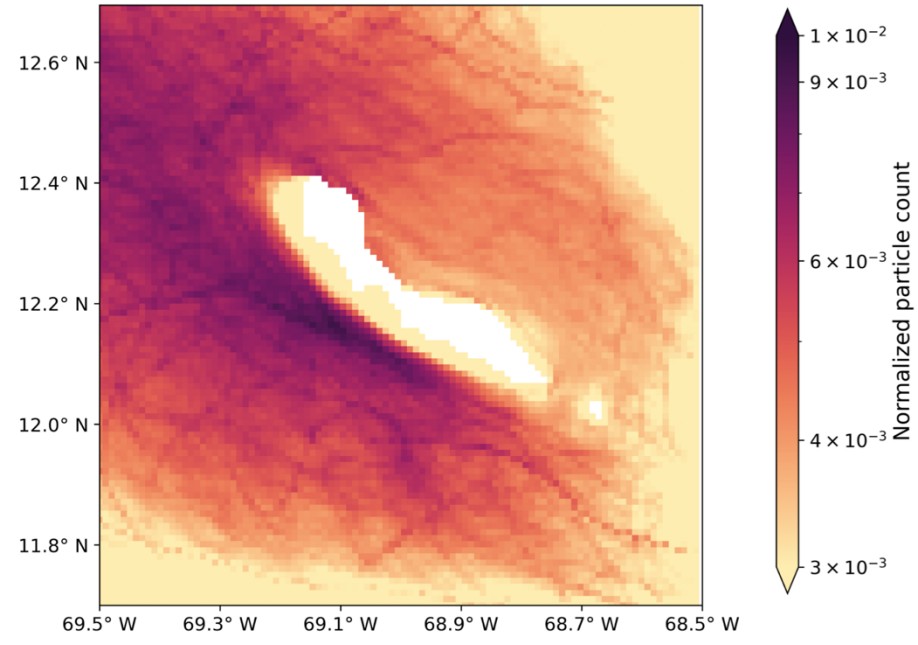

## b) Monthly normalized unique particle count

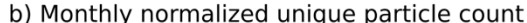

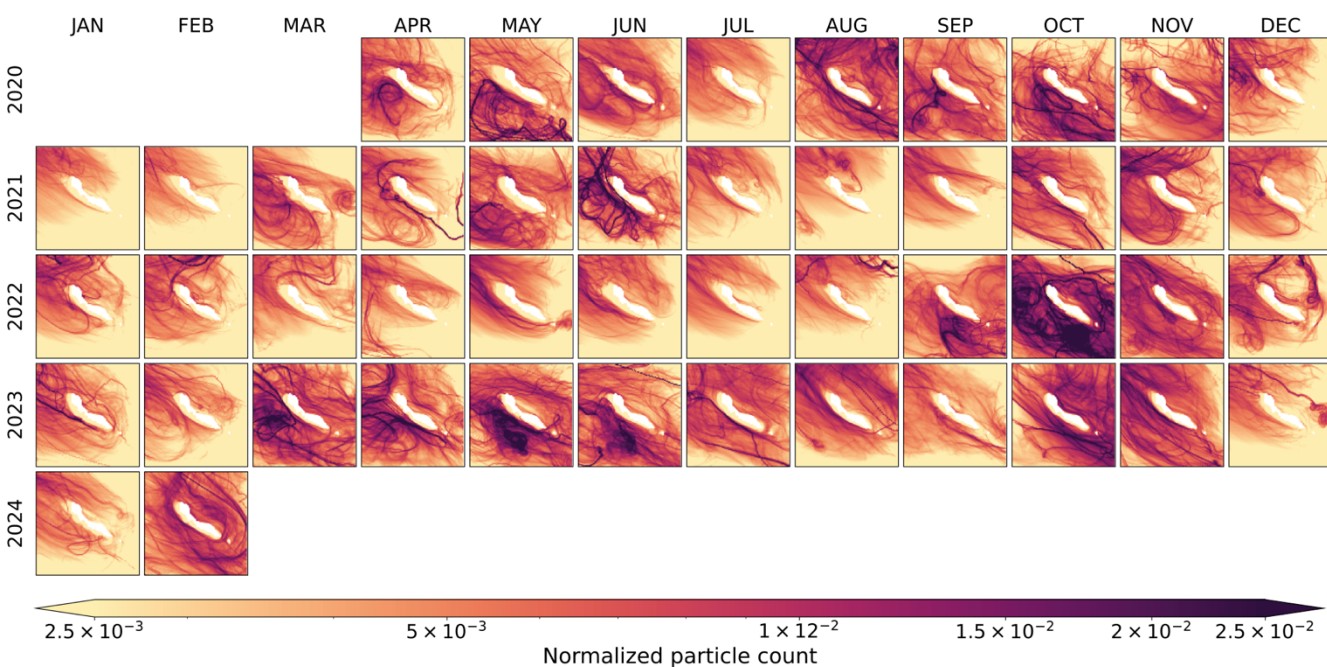

Figure 8: Normalized unique particle counts of particle trajectories around Curaçao, simulated with Scenario 1 method: (a) time-averaged over the entire simulation period and (b) displayed separately for each month of the simulated period.

The monthly analysis of these hotspots (Fig. 8b) reveals monthly and inter-annual variability, highlighting areas with notable differences in particle visitation around Curaçao. Certain months, such as August to November 2020, September to November 2022 and March to November 2023, exhibit elevated particle counts across the entire domain. These periods correspond to lower-energy conditions associated with cyclonic eddies (depicted in Fig. 6b), which result in broader particle distribution across the domain. In other months, higher particle counts are concentrated in the northwestern part of the domain, as a result of the stronger northwest-directed flow regime dominating these months. The disconnection between the southern coastline and the surrounding currents persists throughout nearly all simulated months, with the width of this disconnected coastal area varying based on prevailing hydrodynamic processes each month. This disconnected area is narrowest during cyclonic eddy events, such as those observed in September to November 2022, when particles accumulate extensively around the island and reach both coastlines. Conversely, during strong flushing events characterised by strong northwest-directed currents, particles are pushed further away from the southern coastline, resulting in a greater distance from the land for these particles. During such events, particles are also more likely to leave the domain via the western and northern boundaries. This phenomenon is particularly evident in the months of DJFM, where the surface zonal flow (Fig. 6b) consistently shows a strong westward component throughout all these months. However, in 2024, this pattern is interrupted, as the flow in February lacks the typical strong westward direction.

### 3.3 Intra-island connectivity

The connectivity analysis within the 8 coastal zones of Curaçao is conducted to better understand the interaction between zones (Fig. 9a) and the potential spread of substances, particularly from highly populated areas (zones 3 and 4) with significant coastal pollution to more pristine regions. Figure 9b presents the average connectivity matrix among 8 zones for the entire duration of simulation period, generated with Scenario 2 simulations, demonstrating a notable particle flow towards the northwest. Specifically, zones 3 through 6 exhibit particle transport to their immediate northwest neighbouring zones. This pattern highlights a predominant northwestward particle movement within the region, described in Sect. 3.1. Additionally, each zone exchanges particles with its nearest neighbours on both sides, showing consistent two-way movement throughout the system.

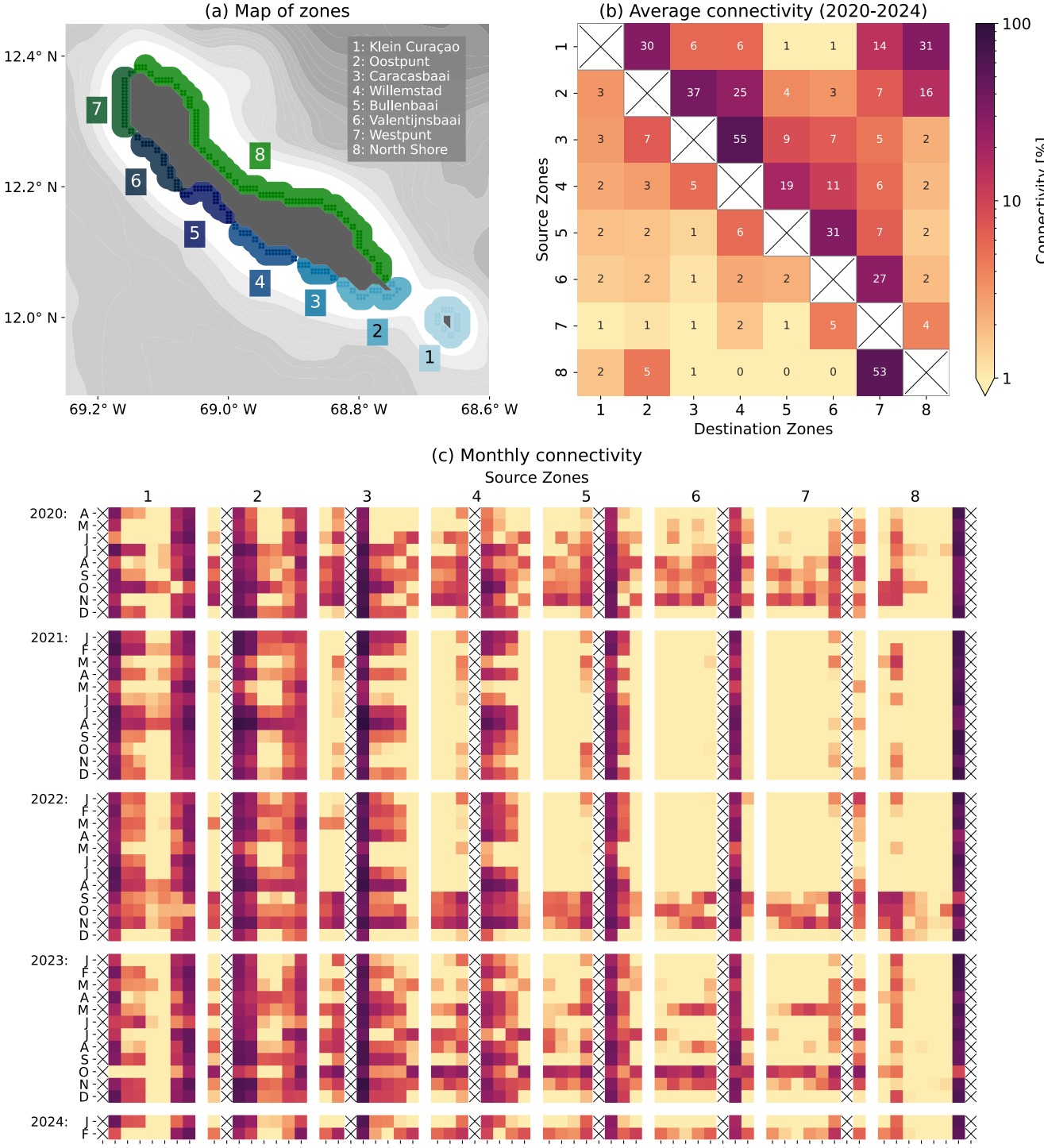

**Figure 9: (a) Particle release locations (dots) for Scenario 2, colour-coded by coastal zones, with coloured destination areas. (b) Connectivity matrix, generated in Scenario 2, showing the average connectivity over the period from April 2020 to February 2024. (c) Monthly connectivity matrix, where rows represent months across the years, and columns indicate connectivity. The wide columns represent the source zones, and each wide column contains 8 narrow columns representing the destination zones, with the destination zone numbers annotated at the bottom. Both (b) and (c) share the same colour bar, representing the percentage of connectivity.**

Zone 7 (Westpunt) experiences significant particle impact, receiving the highest number of particles overall, especially from its neighbouring zones. In contrast, zones 1, 2, and 8 show notably lower influence compared to other zones, particularly receiving fewer particles from the zones with higher population densities (here considered zones 3 and 4), suggesting that these zones may experience lower direct (anthropogenic) impact from in the rest of Curaçao.

Figure 9c shows the monthly variations in connectivity, with source zones represented as wide columns labelled at the top and their corresponding destination zones indicated at the bottom. The northwestward flow of particles dominates across most zones, with the exception of zones 7 (Westpunt) and 8 (North Shore), which have no neighbouring zones to the northwest. A notable period from December 2020 to August 2022 stands out, as connectivity from source zones 6, 7, and 8 is very limited, except to their immediate neighbours. This coincides with a prolonged phase of predominantly northwest-directed surface flow (Fig. 6), which remains strong for most of this period. Interestingly, even during months with weaker flow in this period (e.g., May, June 2021), connectivity from these sources remains minimal. However, during specific months associated with cyclonic eddies – notably August to November 2020, September to November 2022, March to November 2023 and February 2024 (see Fig. 6b) – a distinct southwestward particle movement is also observed. In these months, particles are dispersed more widely, traveling around the island and reaching distant areas such as Klein Curaçao (zone 1). This shows that while the overall trend indicates a northwestward flow, the presence of eddies enhances interactions across the entire island.

**3.4 Coastal connectivity**

The influence of distant locations, such as coastal regions of other islands and the mainland of Venezuela, may also serve as significant sources of substances entering the waters around Curaçao. Figure 10 illustrates connectivity of coastlines within the model domain: Aruba (10b), Bonaire (10c), Venezuelan Islands (10d), and Venezuelan continental coast (10e). The connectivity matrix shows expected minimal connectivity from Aruba to Curaçao, occurring only during five months of the four-year simulation period. These months align with some of the periods of high normalized unique particle counts shown in Fig. 8. However, there are many other months with high particle counts during which no connectivity from Aruba is observed.

In contrast, connectivity between Bonaire and the Venezuelan Islands and Curaçao reveals high particle transport to Curaçao, aligning with the average surface flow patterns depicted in Fig. 6a. Monthly analysis of connectivity between Bonaire and the

Venezuelan Islands towards Curaçao does not exhibit a clear seasonal or inter-annual pattern, suggesting consistent connection throughout the year.

Connectivity from the Venezuelan mainland to Curaçao is relatively low but consistently present across most months, as expected given the extensive coastline. Although the percentage of particles reaching Curaçao is small, this does not diminish the potential influence of these regions as a source of substances. Notably, during nine months of the simulation, no particles from the Venezuelan mainland reached Curaçao. However, these months do not align with any specific seasonal pattern or other observed trends in surface flow or hotspot analysis. This irregularity further emphasizes the complex and variable nature

of oceanic processes in this region. Moreover, since our simulations were designed to terminate early if the computations take too long and only 1% of particles remained in the domain, this primarily occurred in the simulations of the Venezuelan mainland, due to the large number of particles released and the low currents near the mainland. Most of the remaining particles are located near the Venezuelan mainland, particularly along the east coast of the Paraguaná Peninsula.

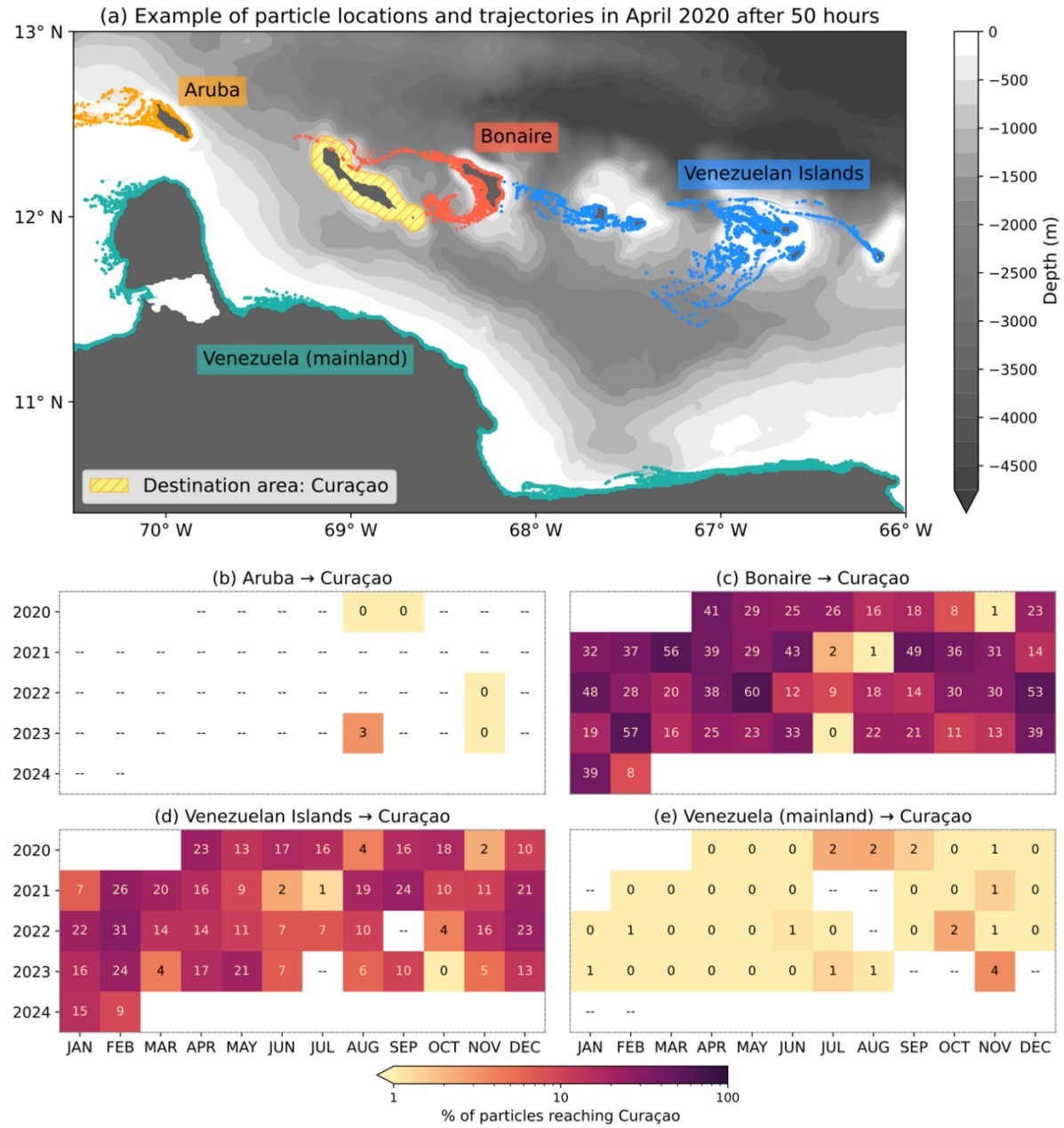

**Figure 10: Connectivity matrix, generated in Scenario 3, for the period April 2020 to February 2024, showing connectivity to Curaçao from (b) Aruba, (c) Bonaire, (d) the Venezuelan Islands, and (e) the Venezuelan coastline. "--" indicates zero connectivity during the release month of the particles, although their trajectories may extend into subsequent months. (a) shows example of particle locations in April 2020, 50 hours after their initial simulation time.**

# 4 Conclusions and discussion

## 4.1 Flow patterns

Monthly analysis of particle trajectories and ocean currents reveals high variability in processes affecting the flow dynamics around Curaçao. Two main processes are observed, the strong northwest-directed Caribbean surface Current and the presence of cyclonic eddies diverting from this main current. These processes dictate hotspots of particles and connectivity between the surrounding coastal areas. While definitive conclusions are limited by the four-year simulation period, the observed seasonal to inter-annual patterns provide strong evidence for the robustness of these processes.

Cyclonic eddies, moving slowly at speeds between 0.12 m s$^{-1}$ (Carton and Chao, 1999) and 0.15 m s$^{-1}$ (Murphy et al., 1999), play a significant role in the seasonal variation of current patterns observed in our model. These eddies, which can persist for several months, are present during the peaks of either the dry or wet season, or both. Previous research has shown that such eddies occur approximately every three months (Carton and Chao, 1999). The interaction between these cyclonic eddies and the predominant northwestward Caribbean Current creates conditions that lead to significant spatial and temporal differences in hotspots of particles, as will be discussed in Sect. 4.2 on particle movement.

The interaction between currents and the island perturbs the strong northwest-directed oceanic flow. This current-island interaction results in generation of eddies in the lee. The strong horizontal divergence leads to significant differences in speed between currents upstream and in the lee of the island. In our study, this effect is particularly notable during periods when the currents are strongly northwest directed. In these periods, vortices form due to the island's influence on flow dynamics, resulting in reduced flow strength in the northwest of Curaçao.

## 4.2 Movement of particles

Understanding the movement of particles is crucial for evaluating how substances (e.g. pollutants and nutrients) are transported around Curaçao. The monthly simulations of hotspots around Curaçao reveal a complex dynamic, particularly along the southern coastline, where a notable disconnection exists between the nearshore currents and those passing further offshore. Particles released nearshore are primarily transported offshore by surface currents. Concurrently, particles released from locations further away rarely reach the coast due to the separation between the nearshore flow and the ambient offshore currents that pass at a distance from the island. This results in a low accumulation of particles near the southern coastline.

During cyclonic eddy events, the situation changes, as the reduced current speeds allow particles to remain closer to the island and for longer periods, potentially bringing them towards the coastal coral reef communities. While the predominant northwestward Caribbean Current generally acts as a flushing mechanism, carrying particles away from the island and reinforcing the disconnection between nearshore and offshore currents, the reduced current speed during cyclonic eddies

narrows the band along the southern coastline associated with this disconnection. As a result, pollutants have a greater chance of reaching coastal coral reefs more persistently and for extended periods. However, the hotspot analysis indicates that this nearshore disconnected band, although narrower, still persists during these events. Similarly, land-derived substances may remain near the coastline for longer due to the decreased current speed during cyclonic eddy dynamics. Despite these conditions, the hotspot analysis indicates high particle counts along the southern coastline of Curaçao only occasionally and not consistently with every cyclonic eddy event (Fig. 8b).

Finally, it is important to note that the particle simulation only considers the surface ocean layer, which we correspond to the top 10-20 m of the water column. While this approach provides a reasonable approximation of average conditions within this depth range, it neglects vertical movements and transport, which are crucial for studying nutrient fluxes. Upwelling, where deep nutrient-rich waters rise to the surface, and downwelling, where land-derived substances sink to deeper layers, play key roles in substance dynamics. However, our Lagrangian particle tracking analysis does not simulate these processes. Incorporating vertical transport into the model will reveal how upwelling processes contribute to nutrient supply and influence coral reef health (e.g. Andrews and Gentien 1982; Leichter and Genovese 2006; Radice et al. 2019; Stuhldreier et al. 2015), while downwelling may reduce neutrally buoyant pollutant concentrations near coastal reefs by removing them from the surface. Investigating the interaction between surface and subsurface processes will provide a more accurate depiction of substance dynamics and their impact on coral ecosystems. Future research should focus on integrating vertical layers into particle tracking modelling and investigating areas where coral reef communities benefit from deep, nutrient-rich waters.

**4.3 Effect on coral reefs**

The interaction of hydrodynamic processes around Curaçao leads to significant variability in the exposure of coral reef communities to particles from different sources over time. This variability is crucial for understanding the stressors that coral reefs face, as exposure to pollutants and nutrients can have detrimental effects on coral health. Coral reefs are highly sensitive to changes in water quality (e.g. Fabricius et al. 2012), particularly to increases in nutrient concentrations and pollutants, which can lead to shifts in community structure and reduced coral cover (e.g. Brown et al. 2017; De'ath and Fabricius. 2010). Our study focuses on passive particles representing pollutants and nutrients, which are positively buoyant and non-degradable. These represent idealized conditions. In reality, many pollutants and nutrients behave beyond these assumptions, such as degradation and interactions with other substances, thereby highlighting the limitations and scope of our approach.

Our findings highlight the varying particle loads on nearshore environments, with the most pronounced differences observed between the northern and southern coastlines of the island. The northern coastline consistently experiences high particle counts throughout the entire simulation period, indicating a persistent exposure to particle loads. In contrast, the southern coastline is protected by the disconnection between nearshore and offshore currents, which limits the accumulation of particles nearshore.

Despite the absence of strong accumulation near the southern coastline, low current velocities in this region need to be considered, particularly during cyclonic eddy events. These low velocities could result in substances from land-based sources remaining near the coast with increasing residence times, as the reduced flow would prevent their dispersion and flushing. However, our particle simulations do not capture these nearshore accumulations.

Another crucial factor is the distinct difference in coral reef distribution between the northern and southern coastlines. The northern coast, which consistently exhibits high particle counts, has relatively low coral reef cover (Waitt Institute (2017) report). This suggests that while the northern coastline is the particle hotspot, these findings do not directly link to the impact on coral reefs, as the reefs are largely absent in this region. On the other hand, the western zones of southern coastline (zones 6 and 7, Fig. 9a) emerge as critical areas of concern for coral reef health. These zones are most likely to be affected by land-derived substances originating from Curaçao, as the prevailing northwestward currents direct the particles towards them.

Furthermore, the connectivity analysis indicates that the areas adjacent to the capital city Willemstad (zone 4, Fig. 9a) are also largely exposed. Urban runoff from this region poses a significant threat to nearby coral reef communities, as pollutants released here are likely to have greater impact on coral reefs. While our study does not explicitly model pollutant sources and concentrations, it is reasonable to expect that areas closer to Willemstad face higher risks compared to more remote areas, such as Klein Curaçao, which is uninhabited and unlikely to contribute significant pollution.

In addition to local land-derived substances, particles from neighbouring coastlines, such as Aruba, Bonaire, and Venezuela, contribute to the overall particle load that reaches Curaçao's coral reef ecosystems. While connectivity from Aruba is minimal, the consistent upstream connection with Bonaire and the Venezuelan islands increases the likelihood of substances from these regions impacting Curaçao's reefs. Although connectivity with the Venezuelan mainland is relatively low, it can have a substantial impact, as Venezuela has significant outflows from rivers, which can carry large quantities of various substances, such as microplastics and microorganisms like bacteria and viruses attached to buoyant debris. Additionally, spill events from oil refineries along the western coast of Paraguaná, Venezuela, are among the documented sources of oil pollution in the region (Croquer et al., 2016). Furthermore, although beyond the scope of this study, it is important to consider the even more distant influences, such as the Amazon and Orinoco rivers, which occasionally reach the southern Caribbean Sea and contribute to the overall substance transport in the region (Coles et al. 2013; Hellweger and Gordon et al. 2002).

## 4.4 Model limitations

While the SCARIBOS model provides valuable insights into ocean currents around Curaçao, there are limitations that should be considered when interpreting the results. The model does not account for wave effects such as Stokes drift, which could significantly influence particle movement in the upper ocean layers (Rühs et al. 2024). In our study area, this effect is expected to be most pronounced along the northern coastline of Curaçao, where wave action is notably stronger. Since waves are

primarily driven by the easterly trade winds and generally flow toward the island, they would further push particles toward the northern coastline. Although our results already indicate the highest accumulation values in this region, including Stokes drift will likely amplify these effects.

The absence of land topography in the model overlooks how Curaçao's terrain, including its highest peak at 372 m, influences local wind patterns and alters surface ocean currents. This effect is particularly significant along the southern coastline. Here, the current's behaviour could be notably different if variations in wind were considered, as the coastal currents change in this region. However, with the 1/100° resolution of SCARIBOS model, capturing these local effects is challenging. The model's ~1 km resolution is insufficient to accurately represent the narrow nearshore bands where these effects are most pronounced. Additionally, the model does not account for locally complex coastal areas on scales finer than 1 km, which can significantly alter local currents and create small, localised hotspots of low current speed and recirculation. These nearshore hotspots require finer resolution models for accurate study.

### 4.5 Implications and future directions

While this study focuses on land-derived substances such as nutrients and pollutants, its methodology can be adapted to study the transport of other particles in the ocean as well. For instance, the model serves as an initial approximation for tracking buoyant (macro)plastic pollution too, which floats on the sea surface and can impact coral health through entanglement and smothering (Lamb et al. 2018; Nama et al. 2023). By taking the advantage of 1/100° resolution SCARIBOS simulation, it is possible to resolve smaller-scale transport pathways and retention zones that would remain unresolved in coarser models. This provides critical insights for predicting how plastic debris accumulates in specific areas, which can directly inform mitigation strategies. Furthermore, understanding surface currents is relevant for studying the movement of coral larvae, which often remain in the upper water column, as the study connects coral populations across coastal areas of the model domain. The detailed connectivity revealed by SCARIBOS allows for a more precise understanding of coral larvae dispersal, which is essential for identifying key areas that maintain genetic diversity and resilience in coral reef ecosystems. The strong connectivity observed within these areas also suggests that coral diseases, such as the recent coral tissue disease observed around Bonaire (Pepe 2024), can reach Curaçao, resulting in another potential threat to local coral reef communities. The timescale for such travel can be as fast as 40 hours when the currents are strong and in the right direction.

SCARIBOS simulations and the Lagrangian connectivity pathways could also be applied to predict pollutant spills and their potential environmental impact. By adapting our methodology, the model could be expanded for operational use by incorporating forecasted boundary and forcing conditions. This would enable more accurate predictions of pollutant transport and support the development of rapid-response tools for environmental risk management.

The study underscores the broader applicability of the results for marine conservation and management strategies in Curaçao. Identifying critical areas – such as regions with coral reefs that are particularly susceptible to high pollution loads – on a regional scale is essential for informing policymakers and stakeholders, enabling them to implement targeted measures such as improving wastewater treatment management, thereby contributing to the protection and sustainability of marine ecosystems on Curaçao.

**Author Contributions:**

VB run the numerical models and Lagrangian particle tracking, conducted the analyses and wrote the manuscript. EvS, FM and PS supervised the work, reviewed the manuscript and provided valuable feedback throughout the process.

**Code and data availability:**

The SCARIBOS dataset is accessible at Bertoncelj (2025a) via https://doi.org/10.25850/nioz/7b.b.7h, providing immediate access to hourly surface currents and water level time series for all grid points of the domain. Full model output is available upon request through the same DOI. Additionally, the SCARIBOS configuration files used to run the model are archived at Bertoncelj (2025b): https://zenodo.org/records/14697794. The scripts for post-processing the model output and re-creating figures, are available in the GitHub repository: https://github.com/OceanParcels/SCARIBOS_ConnectivityCuracao. The scripts for reproducing all Parcels simulations, analysing them and re-creating the figures can also be found in this GitHub repository. The ADCP data for model assessment is available at Bertoncelj (2024): https://doi.org/10.25850/nioz/7b.b.xh. The bathymetry, collected with RV *Pelagia*, is integrated in EMODnet Digital Bathymetry product (EMODnet Bathymetry Consortium, 2022), and can be downloaded at: https://doi.org/10.12770/ff3aff8a-cff1-44a3-a2c8-1910bf109f85.

**Competing Interests:**

At least one of the (co-)authors is a member of the editorial board of Ocean Science.

**Acknowledgements**

This publication is part of the project "Land, Sea, and Society: Linking terrestrial pollutants and inputs to nearshore coral reef growth to identify novel conservation options for the Dutch Caribbean (SEALINK)" with project number NWOCA.2019.003 of the research program "Caribbean Research: a Multidisciplinary Approach" which is (partly) financed by the Dutch Research Council (NWO). We sincerely acknowledge the crew of RV *Pelagia* during the 64PE500 and 64PE529 expeditions for their work in collecting bathymetry data and supporting the acquisition of ADCP data. Furthermore, we sincerely thank the two

anonymous reviewers for their valuable comments and suggestions, which significantly enhanced the clarity, precision, and overall quality of the manuscript.

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
