# Peer review of "Flow patterns, hotspots and connectivity of land-derived substances at the sea surface of Curaçao in the Southern Caribbean"

_EGUsphere, 2024_

## Referee Comment (RC1)

**Summary**

Bertoncelj *et al.* describe a 1/100 degree configuration of the CROCO ocean model for the Netherlands Antilles. Using four years of model surface current output combined with Lagrangian particle tracking experiments, they focus on the island of Curaçao and investigate (i) accumulation 'hotspots', (ii) intra-island connectivity, and (iii) the potential for nearby islands and the Venezuelan mainland to act as sources of pollutants for Curaçao. I enjoyed reading the manuscript and, assuming the model output is indeed made publicly available prior to publication, I imagine that these data will be valuable more broadly, for a part of the ocean that appears to be somewhat data sparse. I was particularly impressed with the quality of many of the figures. I recommend the manuscript for publication, but would ask the authors to consider the suggestions below, which can be considered as major revisions. These suggestions mainly relate to (i) more rigorous assessment of model hydrodynamics, and (ii) the clarity of some analyses/discussion, particularly relating to scenario 1.

**General comments**

1. The study would benefit from some more rigorous assessment of the hydrodynamic model. The authors show that SCARIBOS has good performance for tidal water levels (at least at one location), but tides presumably only have a minor effect on particle dispersal around Curaçao compared to the prevailing currents and eddies (and reproducing water levels associated with tides does not necessarily mean tidal currents are reproduced well). The comparison in Figure 2 with ADCP data is very good, but this is one snapshot in time. There is some further assessment of the model in section 4.1, but this is vague and qualitative. The manuscript describes SCARIBOS as a whole (which covers a much broader domain than just Curaçao), but the hydrodynamics are only assessed immediately around Curaçao. I would make the following suggestions:

   a. Surface currents (mean state, and ideally variability) should be assessed for the *entire domain* against an (ideally) observational product, e.g. GLOBCURRENT (and this should be done quantitatively and/or graphically, not just described in the text). Remote-sensing surface current products for the region will have a much coarser resolution than SCARIBOS, and will of course not resolve interactions with the islands, but will at least allow the reader to be satisfied that SCARIBOS reasonably reproduces the large-scale current systems and their variability (I did this very quickly by eye and it looks like SCARIBOS does quite well).

   b. Given the importance of cyclonic eddies for the dispersal dynamics described in the manuscript, quantitative comparison of eddies simulated by the model to observations would also be useful, if possible, particularly since the ADCP observations were during a non-eddying time. The manuscript cites several papers that discuss eddy dynamics in the region, so I would ask the authors to consider whether there are observational data they could use for this assessment.

   c. The authors could also consider using Global Drifter Program drifter trajectories – if there are sufficient observations in this region – to compare true surface dispersal pathways (in a statistical sense) to simulated trajectories. This would be useful in the context of inter-island connectivity for Scenario 3.

2. I have a few concerns about the hotspot analysis (scenario 1):

   a. I'm finding it difficult to see the 'big picture' with this analysis, probably because there is so much variability in Figure 7 – perhaps the authors have tried this already, but I wonder if plotting a time-average (e.g. across the 'normal' state and 'eddy-dominated' states) might make spatial patterns clearer here.

   b. L47 defines hotspots as "areas where land-derived substances spend considerably more time than in other areas" – however, particles were seeded primarily in ocean grid cells, not coastal grid cells. If hotspots are genuinely supposed to reflect the fate of land-derived substances, particles should only be released from coastal cells (as was the case in Scenario 2). If the analysis is instead intended to identify zones of

accumulation more broadly, the decision to release particles in a 1x1 degree square seems arbitrary, and I would be interested to know how sensitive the PDFs in Fig. 7 are to this decision (e.g. does the described accumulation of particles NW of Curaçao persist if particles are seeded over a broader area?). I also wonder if computing the average residence time of particles might be a useful way of identifying hotspots.

    c. The manuscript discusses hotspots in the context of the Island Mass Effect, evidenced by higher particle densities NW of Curaçao. This may be true, but I wonder if there is physical evidence from the model output (e.g. evidence of convergent surface currents in the lee of the island) that would further support this being due to the IME, as opposed to just being a consequence of NW-ward currents.

3. I was interested in the Waitt Institute report that was cited in the manuscript, which appears to have various data (e.g. sewage indicators, trash accumulation indices, infrastructure density) that could be directly useful in this study – e.g. weighting coastal particles in Scenario 2 by infrastructure density to weight the connectivity matrix in fig. 8 since some parts of the coast are more likely to generate pollutants than others, or testing whether trash accumulation hotspots agree with predicted hotspot locations. I wonder if these data could be useful for model assessment?

**Specific comments**

| Line | Comment |
| --- | --- |
| - | The maps in this manuscript (particularly Figure 1) appear to have been exported in a vector format, but this has made them very large (the size of the manuscript is 17MB) and causes lags when opening the file in a web browser. I would recommend exporting these maps in PNG format instead. |
| 43 | This does not undermine the purpose of this study, but in the interest of accuracy/fairness I would consider citing the NCOM AmSeas model (https://www.ncei.noaa.gov/products/weather-climate-models/fnmoc-regional-navy-coastal-ocean), which covers Curaçao at 1/30 degree resolution. |
| 51 | Here and throughout (particularly section 2.3), I would use the word "assess" rather than "validate". Although most readers will know what you mean, the word "validate" in this context is arguably incorrect (e.g. Konikow & Bredehoeft, 1992). |
| 55 | Here and throughout, I would suggest being more specific about *which* substances the study is attempting to model (assumptions of positive buoyancy, no degradation, etc. will of course only be relevant for certain types of substances). This is particularly important in the context of section 4.3, as many of the pollutants that affect corals are not neutrally buoyant (e.g. sediment) and/or non-conserved (e.g. many nutrients). |
| 59 | Here and throughout, avoid the word "fine" in the context of resolution: it is subjective and context-dependent, just write the actual resolution. |
| Fig. 1 | This is a good figure, a couple of minor comments:
- The contour increments in the colour map are slightly inconsistent with the tick labels in the colour bar (this is also the case for fig. 4).
- The brown land polygons are slightly inconsistent (higher resolution?) with the grey coastline |
| 119 | I assume the vertical resolution is finer at the surface – if this is the case, I would specify it (and possibly give the range of values of the thickness of the upper layer). |
| 120 | Please justify how it was determined that 4 months was a sufficient spin-up duration. |
| 121 | How frequently were currents saved (hourly?), and were these snapshots or averages? What was the frequency of current data used in Parcels? |
| 123 | I assume that smoothing was performed on the bathymetry (as I believe is standard for ROMS/CROCO preprocessing) since very steep slopes can cause stability issues. If this is the case, I think this should be mentioned in the methods. |
| 124 | The comment on "adjusting land grid cells" is very vague – I would briefly add the reason for these adjustments (presumably because the land-sea mask generated from the bathymetry is inconsistent with the true coastline). |
| 134 | What is the source of the river discharge data (and is this based on monthly averages/monthly climatology…)? |
| 189 | Given the relatively limited role of tides in setting marine dispersal, and the fact that the ADCP comparison is based on one snapshot, I do not think we can conclude that "SCARIBOS accurately simulates surface-level dynamics, *making it a reliable tool for tracking surface currents*" on the basis of the evidence presented so far (see general point 1). |

| | |
|---|---|
| 204 | How did you determine the number of particles to be released? Was there a sensitivity analysis? |
| 204 | Is there a reason why particles were released every 24 hours for Scenario 1, but every 12 hours for Scenario 2? I don't think this is an issue, just seems a bit odd that different release frequencies were used. |
| 256 | This mention of El Niño is a bit random – is it relevant to the local hydrodynamics that it's an El Niño year? This is again mentioned in line 368. |
| Fig. 5 | I really like these figures showing the monthly state of the model, I think they are very effective. |
| 267 | Have there been hydrographic surveys in the region (or anything else) that supports this vertical velocity profile and the countercurrent? |
| 271 | I am not sure about tying this in with the AAIW... even if there is a signature of AAIW here, I would not have expected this to be relevant to the dynamics, which is the topic of this manuscript. |
| Fig. 7 | I have a few comments about this figure:

- I would consider either (i) using a coarser binning grid, (ii) having more frequent particle releases, or (iii) outputting particle locations more frequently, because many of these sub-panels have a checkerboard pattern in the PDF. Using a coarser binning grid may also make the figure easier to interpret (by smoothing out some of the filamentation).
- Have you tried producing a time-mean (or seasonal, or non-eddy/eddy mean PDF)? Because there is so much variability, I am finding it difficult to see the 'big picture' from this figure.
- Zooming into the sub-panels, there is something going on in the background of Curaçao, see below (is the particle distribution being plotted on top of a raster outline of Curaçao)? I would remove this and keep the background fully white, otherwise this could be mistaken as representing data (high particle density).

[Figure]
 |

| | |
|---|---|
| Fig. 8 | Panel (a): because the changes in colour between some neighbouring areas are quite subtle, it may be helpful to readers to add numbered labels on the map next to each zone (this would also make it easier to quickly cross-reference between panel (a), and panels (b) and (c))
Panel (c): This is a really cool figure! I wonder if it might be useful to mark (e.g. with horizontal dashed lines) months associated with the passing of major eddies. |
| 329 | However, most of the particles arriving at Zone 7 come from Zone 8 (and to a lesser extent, Zone 6), which are not highly populated. I understand that this study is not attempting to model any one particular pollutant (and therefore does not use a specific input function) but I do think it is important to qualify that, although Zone 7 may receive the most particles, this does not necessarily mean it will receive the greatest pollution burden. A similar point is made in the paragraph beginning on line 438. |
| 340 | Analyses for Scenario 2 assume that substances of interest have a lifespan of 30 days, which seems quite arbitrary. If you think this goes beyond the scope of this study then I accept that, but it would be interesting to know how sensitive Figure 8 is to particle lifespan. |
| Fig. 9 | This could mess up the layout, but it might be useful to have a reference map (similar to fig. 8) to remind the reader of where these different sources are relative to Curaçao. |
| 382 | I am struggling to understand the point being made in this paragraph – can you relate this discussion of the IME to your results? Does the model predict large-scale downwelling in the lee of the island? Does the model predict upwelling (and divergence) along the south coast of Curaçao where there appears to be unidirectional offshore transport in surface waters? |
| 400 | I am not sure I understand why this paragraph is referring to the sticky water effect – my basic understanding (and as stated in the text) is that the 'sticky water' effect specifically refers to currents that are *retentive* but (as discussed in the previous paragraph), currents around the south coast of Curaçao instead appear to be highly dispersive – even though there are some similarities in the physical mechanism, i.e. current diversion around an obstacle. I understand the point being made, but is 'sticky water' the right term to describe it? Likewise in lines 433 and 467. |
| 409 | Similar to the point on line 382, is there evidence (e.g. vertical velocities diagnosed from the model) that prove this is due to the IME? As mentioned in my comment on Fig. 7, I think this might be easier to see with a multi-year seasonal mean. |
| 419 | Change "reduce pollutant concentrations" to "reduce neutrally buoyant pollutant concentrations", since downwelling will *increase* the concentration of positively buoyant pollutants. |
| 442 | It might be useful to mark the location of major settlements on figure 4(b) and/or 8(a). |
| 444 | I am confused by the point being made here. What is meant by "the limited distance travelled by substances such as pollutants" – does that refer to their degradation timescale in the ocean? Surely that depends on the type of pollutant and, regardless, the results in this study still suggest that the south coast probably has low pollutant retention? Similarly, I don't understand the claim that there is "reduced dilution" (L448). Figure 7 shows that particles that start on the south coast quickly move away, and are not replaced by new particles. That sounds like dilution to me. |
| 453 | "Various substances" - I would suggest giving examples of substances carried by Venezuelan rivers that could be modelled by the approach taken in this study (i.e. positively buoyant, lifespan of longer than a month). |
| 478 | Coral larvae are not always positively buoyant – their buoyancy declines with age (Szmant & Meadows, 2006). As they mature and gain vertical swimming ability, they have some control over their position in the water column, which varies considerably by species (e.g. Mulla et al., 2021; Tay et al., 2011). The point you are making here is fine, but I'd change "…which are also buoyant" to "…which often remain in the upper water column" (or similar). |
| 479 | Since there was no limit on the particle lifespan in Scenario 3, I think you need to quantify the connectivity timescale between Bonaire and Curaçao before you make this claim. I am not sure how coral disease is transmitted, but I assume the pathogens and/or vectors have a limited lifespan in the water column? |
| 482 | I would restate in this sentence which specific areas you consider to be "critical areas" – at the moment, this is a very vague and generic statement. |
| 493 | The configuration files should probably be archived on Zenodo or similar prior to publication, since GitHub is not a permanent repository. |

**Technical comments**

| Line | Comment |
| --- | --- |
| 36 | Would rephrase to "Coral reefs are not just impacted by local sources of pollution, but also by broader environmental changes and anthropogenic activities" (otherwise it sounds like "broader environmental changes and anthropogenic activities" are an impact on coral reefs, rather than a source of impact). |
| Fig. 6 | Change B) to (b) in the panel label. |
| 312 | Would change "western" to "westward" |
| 316 | This was mentioned earlier, but I would remind the reader at this point where the highly populated areas on Curaçao are. |
| 344 | Change 9A, 9B etc. to lower case. |
| 348 | Change "a strong signal" to "high particle transport" (or similar). |
| 451 | Consider changing "connection" to "upstream connection" |
| 475 | These sentences ("While this study focuses on… pollutants, its methodology can be adapted… for tracking plastic debris") makes it sound like plastic debris is not a pollutant. |

**References**

Konikow, L. F., & Bredehoeft, J. D. (1992). Ground-water models cannot be validated. *Advances in Water Resources*, *15*(1), 75–83. https://doi.org/10.1016/0309-1708(92)90033-X

Mulla, A. J., Takahashi, C. L. S., & Nozawa, Y. (2021). Photo-movement of coral larvae influences vertical positioning in the ocean. *Coral Reefs*. https://doi.org/10.1007/s00338-021-02141-7

Szmant, A. M., & Meadows, M. G. (2006). Developmental changes in coral larval buoyancy and vertical swimming behavior: Implications for dispersal and connectivity. *Proceedings of the 10th International Coral Reef Symposium*, *1*, 431–437.

Tay, Y. C., Guest, J. R., Chou, L. M., & Todd, P. A. (2011). Vertical distribution and settlement competencies in broadcast spawning coral larvae: Implications for dispersal models. *Journal of Experimental Marine Biology and Ecology*, *409*(1–2), 324–330. https://doi.org/10.1016/j.jembe.2011.09.013

---

## Referee Comment (RC2)

**Summary**

Bertoncelj et al. introduce SCARIBOS, a regional configuration of the CROCO ocean model for the South CARIBbean Ocean System with a kilometre-scale horizontal resolution of 1/100 degree. The authors use four years (2020-2024) of surface velocity fields output by the model to undertake three Lagrangian particle tracking experiments using the OceanParcels framework to explore the surface connectivity of flows surrounding the island of Curaçao. The Lagrangian experiments allow the authors to investigate (a) potential hotspots of marine pollutants around Curaçao, (b) intra-island coastal connectivity, and (c) the connectivity between the Curaçao coast and the neighbouring Aruba, Bonaire, Venezuelan islands, and the Venezuelan mainland. The authors should be commended for the Figures, especially those relating to the Lagrangian analysis, which convey the central findings both clearly and creatively. The manuscript is generally well written, although a more comprehensive description of the numerical modelling approach (including model validation), further exploration of the role of mesoscale eddies in driving surface connectivity, and an improved discussion on the limitations and wider relevance of the findings is needed. I would recommend the manuscript for publication subject to major revisions addressing the comments made below and the excellent suggestions made by Reviewer #1.

**General Comments**

- ***The current title:*** 'Flow patterns, hotspots and connectivity of land-derived substances at the sea surface of Curaçao in the Southern Caribbean' feels like a description of what the author's set-out to investigate in this study but does not give any indication of the main findings or conclusions drawn from the analysis. Although the title is perfectly acceptable in its current form, I wonder whether it could improved to highlight the findings most relevant for stakeholders / policymakers.

- ***Model description and validation:*** Given that this is the first documentation of the SCARIBOS simulation and it is likely this will be used by other studies in the future, further details are needed to improve the model description, including highlighting the absence of Stokes Drift, stating whether a current feedback parameterisation is implemented, commenting on key parameterisation choices of ocean physics (e.g. in the surface mixed layer), and including the frequency at which velocity and tracer fields are output (this will be especially relevant for future studies). Moreover, as highlighted by Reviewer #1, the validation of the simulation is currently insufficient to robustly conclude that it accurately represents the circulation of the Caribbean Sea (see specific comments below). Moving forward, I would suggest broadening the validation beyond Curaçao to the wider Caribbean Sea domain and to use several sources of observations,

including surface drifters deployed in the region during the simulated period (2020-2024).

- ***Extracting maximum value from the 1/100 degree SCARIBOS simulation***: the Lagrangian analyses presented in the study are all well motivated and provide new insights into surface connectivity in the region. However, on reaching the conclusions of the manuscript, I did not feel that the full potential of this state-of-the-science simulation had been extracted. Firstly, it would be valuable to know if the insights presented are critically dependent on resolving flow structures at 1/100 degree or could similar conclusions be drawn using a much lower resolution. One powerful way to illustrate this would be to spatially coarsen the surface velocity field and repeat one or more of the Scenarios to examine how this impacts marine connectivity. A natural caveat here would be that the coarsened velocity field still originates from one which explicitly represented (sub)mesoscale dynamics, however, identifying the critical threshold of horizontal resolution required to represent connectivity in this region would be an incredibly valuable contribution of this work and would extend beyond the present application to Curaçao.

  Similarly, I think the authors could better exploit the eddy-resolving nature of the simulation to further investigate the role of mesoscale eddies beyond the primarily qualitative descriptions included currently. A valuable example of such an approach is Roach and Speer (2019) - https://doi.org/10.1029/2018JC014845 - which identified the timescales of variability in the flow field which are responsible for the connectivity between the Ross Gyre and the Antarctic Circumpolar Current by coarsening their 5-day mean velocity field to 90-day means and a single time-mean field. This allowed them to separate the connectivity associated with high-frequency (e.g., mesoscale eddies) and low-frequency (e.g., seasonal variability) variability from the time-mean flow. Given SCARIBOS explicitly resolves the (sub)mesoscale, using such a time-coarsening approach in this study could provide new insights into the role of high-frequency flow features in establishing the surface connectivity around Curaçao. This may also offer further retrospective justification for the use of a 1/100 simulation and underscore the value of such regional models for informing policy making and marine planning.

**Specific Comments**

**Abstract**

Lines 11-12: Suggest replacing ', as these substances…' with 'since these substances can be transported towards reef sites by ocean currents' given that its already implied that the substance has entered the ocean.

Lines 13-16: Suggest combining the two sentences beginning with 'SCARIBOS, a fine-resolution...' and 'SCARIBOS covers the...' to reduce repetition. Then 'Furthermore,' can be dropped in the following sentence as it is not needed.

Lines 19-21: Suggest condensing these two sentences to be less ambiguous. For example: 'Our results reveal two dominant processes influencing the hotspot locations of positively buoyant substances...'

Line 24: As a non-domain expert on marine pollutants, I was surprised to see justification of this work as providing valuable information for marine conservation and environmental management at the end of the Abstract. The need for kilometre-scale modelling of ocean connectivity to inform stakeholders' decision-making struck me as an important motivation for pursuing the study alongside the current 'problem statement' highlighting the general decline of coral reef communities.

**Introduction**

Line 28: Suggest being more specific on the rates of coral reef decline, by how much has this changed already? Is this accelerating? This would further strengthen the motivation for the study.

Lines 30-32: Suggest combining these two sentences: 'are susceptible to accumulating pollutants, bacteria and viruses originating from urban areas, ...'

Lines 34-35: As a non-domain expert, it would be interesting to know how the threat posed by the accumulation of marine pollutants and harmful biological substances around coral reefs compares to other threats, such as marine heat waves and ocean acidification. Why is this threat especially worth investigating?

Lines 36-46: Suggest revising or restructuring this paragraph as it's currently difficult to follow: it begins with a recognition that remote sources of marine pollutants are important for coral reefs, then proceeds to discuss why existing numerical model simulations are insufficient to represent the local ocean dynamics of the Caribbean, and concludes that developing a SCARIBOS model is the answer to this challenge. I wondered whether an alternative framing of this paragraph could be to highlight that in almost all ocean general circulation models, most notably the CMIP6 ensemble, Caribbean small island states are (at best) represented by a single grid cell or not at all, hence we are currently not in a position to translate global nor regional scale insights to local communities (who may be significantly impacted by unresolved processes). Thus, developing fine-resolution regional configurations like SCARIBOS provides a means to represent these regions more accurately and inform marine conservation and environmental management efforts.

Lines 44-47: Suggest refining this conclusion to be more precise. The preceding text highlights that there is only limited high-resolution ocean model data for the Caribbean

Sea, but why does it follow that we need a high-resolution model of Curaçao specifically. Is the coral reef environment here particular at risk or subject to emerging risks? Or are the ocean dynamics here representative of the wider region, such that the insights drawn from this study are applicable elsewhere?

Line 49: Suggest modifying to 'our research investigates **the** dominant surface ocean current patterns and substance transport **pathways** around Curaçao...'

Line 50: Do you mean monthly to inter-annual **variability** rather than environmental changes? I found this to be ambiguous as ocean variability is discussed later in the text.

Line 51: Suggest using vertical **extent** here and elsewhere rather than reach in this context.

Lines 59-67: Not entirely sure this paragraph summarising the methodology is necessary on reading the full text since the description of the model and methods directly follows the introduction. As a compromise, it could be condensed to focus on the development of SCARIBOS as an answer to the research question which is nicely presented in the previous paragraph.

**Methods**

Figure 1: Suggest here and elsewhere ensuring that the colourbar ticks and contour levels are aligned to make it clear that the sea floor in the regions shown in white, for example, lies between 0-500 m depth etc.

Line 92: More appropriate, given the geographical location of this study, to refer to the Atlantic Meridional Overturning Circulation since you are referring to the warm upper limb waters feeding the Florida Straits.

Lines 120-121: Is this really a sufficient model spin-up time to ensure that the Lagrangian experiments are not capturing the ongoing adjustment of the mean flow and eddy kinetic energy fields? How was this determined? Further details should be included here.

Lines 122-125: What is the spatial resolution of the bathymetry product linearly interpolated from? What manual adjustments were made to the bathymetry? Further details on how this was undertaken and to what extent this (presumably) improved the resulting flow structures around the island should be included.

Lines 129-131: Is the interpolation scheme used to downscale the GLORYS12V1 velocity and tracer fields bilinear or conservative (conservative-normed)?

Line 132: When applying the surface atmospheric forcing does SCARIBOS account for the current feedback to the atmosphere (CFB), which contributes to the oceanic circulation by damping mesoscale eddies. (i.e., Does CROCCO account for the fact that the ERA5 surface wind stress field acting on the ocean has already 'felt' the surface

ocean currents and hence simulations forced without CFB overestimate the mean circulation and the mesoscale activity). An excellent discussion on the use of reanalysis winds to force ocean models in provided in Section 6 of Renault et al., (2020) - https://doi.org/10.1029/2019MS001715.

Lines 134-135: At what frequency are the 2-D (surface) and 3-D velocity and tracer fields output from the SCARBIOS model? Are daily mean fields being used as the inputs to OceanParcels? This is currently unclear and should be added to both the model description and the description of the Lagrangian experiments undertaken in this study.

Lines 189-193: Unfortunately, I do not think that sufficient validation has been undertaken to justify the conclusion that 'SCARIBOS accurately simulates surface-level dynamics'. In this section, the model has been shown to reproduce the sea level timeseries at a single location and time-average currents agree qualitatively with limited observations in magnitude and direction. Reviewer #1 has made a number of excellent suggestions on ways to improve this validation, which I will not repeat here. I strongly support the use of surface drifters to validate the surface flow field. More broadly, I also think that the authors should comment (either in the methods or discussion section) on the somewhat philosophical challenge of undertaking simulations at kilometre-scale resolution in regions where observations are sparse – how do we know what good looks like? This also relates to my more general comment; I think it would be valuable for the authors to consider what is the minimum horizontal resolution needed to investigate connectivity in the Caribbean Sea (see general comments above).

Line 197: Suggest acknowledging here that the velocity fields do not include Stokes Drift rather than leaving this until the Discussion.

Lines 204-205: To be clear, does this equate to releasing a single particle in each 1/100 grid cell? How sensitive are the results of Scenario 1 to these initial conditions given that (sub)mesoscale turbulence is explicitly resolved in this simulation? Conceivably, given how chaotic the underlying velocity field is, a small difference in the initial position of a particle could result in a very different final position following 30 days of advection. An insightful discussion of the chaotic behaviour of Lagrangian trajectories is presented (albeit applied to ocean ventilation in a much coarser OGCM) in MacGilchrist et al. (2017) - https://doi.org/10.1002/2017JC012875.

Lines 205-206: The statement: 'The internal particle simulation timestep is set to 5 min and trajectories are archived every hour' is repeated for all three experiments. Suggest outlining the common features of the three scenarios in a final paragraph to reduce repetition.

Lines 209-210: Did you also consider calculating Lagrangian PDFs by counting the number of unique entries into each given grid cell normalised by the total number of particles (i.e, calculating the likelihood that any given particle will enter a grid cell at

least once during its lifetime)? In my experience, this can improve the clarity of Lagrangian PDF plots where recirculation features are dominant and better illustrate the net flow pathways.

Lines 215-216: Why are particles released every 12 hours in Scenario 2 compared with every 24 hours in Scenario 1? Is the initial time of release important relative to the diurnal cycle of atmospheric forcing, what time each day does this take place? It would be helpful for the authors to comment more on these uncertainties.

Lines 225-226: Why are particle released two coastal grid cells away? Presumably, beaching of particles could be a problem when using a numerical time-stepping scheme to determine the trajectories?

Lines 236-237: Suggest adding a brief description of the locations of the remaining 1% of particles in Scenario 3 which do not leave the domain. Are these locations consistent between particle releases? If so, would regions of high particle persistence be particularly concerning for marine pollution and environmental management?

Figure 4: Excellent Figure, the authors have done a great job of visualising the differences between the Scenarios. Suggested modification to the final line of the Figure caption: 'The destination area highlighted around Curaçao represents the region within which particles are tagged as reaching the Curaçao coast.'

**Results**

Lines 250-257: In Figure 5b, interannual variability appears to dominate over seasonality, so I would suggest caution not to overinterpret monthly behaviour based on four years of surface velocity data. Caveating the discussion by highlighting the limited number of months available to sample (4 instances each) would be one approach.

Lines 263-266: Why is the analysis restricted to a single meridional cross section? A comparison of the 2-dimensional (longitude-latitude) flow field at various depths would properly account for the spatial dependence of the flow and to make the conclusion that the surface velocity field is representative of the upper ocean flow field more robust.

Lines 270-272: Are the T-S properties of the westward current consistent with AAIW at these latitudes? This would strongly support the inference.

Lines 273-280: This paragraph is quite confusing. The opening sentences largely repeat the findings above and third sentence seems to preface the say that the flow field is highly variable. I would also recommend removing 'observed' on Line 277 since (I think) you are still referring to the output of the SCARIBOS model here?

Lines 289-290: This concluding sentence feels slightly disjointed from the preceding text, which is a very nice synthesis. Perhaps, the component of the discussion that is

missing is: is it reasonable to assume that the vertical motion of marine substances limited to the upper 10-20m which the surface velocity field is representative of?

Line 295: Suggest adding 'reveals significant monthly and inter-annual variability'. More generally, it would be interesting to assess statistically whether the variability seen between monthly release maps is stochastic versus seasonal-interannual in nature. A similarity metric, such as the Fraction of Unexplained Variance (FUV) could be used to compare months and assess how similar any given month is to its monthly climatology (e.g., what fraction of the PDF shown for April 2023 can be explained by the April (2020-2023) average).

Figure 8: As a non-domain expert, I found the large number of connectivity matrices in (c) to be difficult to interpret and, in contrast to the other Figures presented in the manuscript, to be the least effective at highlighting the key result of this Scenario. Two possible suggestions, which the authors are fully entitled to disregard, would be to replace the Source Zone numbers with geographical names as in (a) colouring font according to their location, and either masking or recolouring the 100% connectivity boxes (this value is known since particles are released here, but is the boldest feature in every subplot).

**Discussion and Conclusions**

Lines 370-372: This sentence could be clearer, suggest modifying to: 'There is {broad/strong/good} agreement between the surface current vectors simulated by SCARIBOS and those estimated from Lagrangian surface drifters in the Caribbean Sea between 1989 and 2003 (Richardson, 2005).'

Lines 369-375: Much of this discussion would be better placed in the methodology section validating the SCARBIOS model. This would ensure readers have greater confidence in the simulation's ability to represent the circulation in the Caribbean Sea before it is applied in the Lagrangian analysis, rather than discussing this retrospectively.

Lines 376-381: This brief discussion on cyclonic eddies is interesting, but I feel more could have been done in the Results to explore this (see earlier general comments on temporal coarsening), including coarsening the flow field in time to extract the signature of high-frequency flow components on the connectivity and persistence of marine pollution around Curaçao.

Lines 435-437: This raises an interesting discussion point on the residence times of particles around the coast, however, residence times were not addressed in the Results section. Was this intentional and based on a supplementary analysis of the particle residence times? It would be interesting (perhaps in future work) to combine the findings on the connectivity of positively buoyant marine substances with their

residence timescales in coastal regions around Curaçao, since a highly connected reef with a low flushing (high residence) time scale would surely be more susceptible to marine pollutants.

Line 459-460: The absence of wave effects, including Stokes Drift, should be commented on in the methods section as it's an important limitation of the surface velocity field used in the Lagrangian analysis.

Lines 474-484: This concluding section on 'Implications and future directions' could be improved by emphasising the value of the SCARIBOS 1/100 simulation – what have we learnt with this model which is not attainable at lower resolution – and identifying several future questions which will directly inform policymakers and stakeholders. Currently, plastic debris, coral larvae and marine pollution are discussed collectively, but it would be interesting to know how SCARIBOS outputs could be used in each case, thereby underscoring its long-term value as both a scientific and societal resource. For example, could SCARBIOS be used to predict (or train an ML model to predict) pollutant spills and the resulting environmental impacts?

---

## Author Comment (AC1)

**1 Summary**

Bertoncelj *et al.* describe a 1/100 degree configuration of the CROCO ocean model for the Netherlands Antilles. Using four years of model surface current output combined with Lagrangian particle tracking experiments, they focus on the island of Curaçao and investigate (i) accumulation 'hotspots', (ii) intraisland connectivity, and (iii) the potential for nearby islands and the Venezuelan mainland to act as sources of pollutants for Curaçao. I enjoyed reading the manuscript and, assuming the model output is indeed made publicly available prior to publication, I imagine that these data will be valuable more broadly, for a part of the ocean that appears to be somewhat data sparse. I was particularly impressed with the quality of many of the figures. I recommend the manuscript for publication, but would ask the authors to consider the suggestions below, which can be considered as major revisions. These suggestions mainly relate to (i) more rigorous assessment of model hydrodynamics, and (ii) the clarity of some analyses/discussion, particularly relating to scenario 1.

We thank the reviewer for their detailed and very useful comments and suggestions. Below, we respond to each comment in detail and address them accordingly. Line numbers mentioned in the text correspond to line numbers in the revised manuscript.

**General comments**

1. The study would benefit from some more rigorous assessment of the hydrodynamic model. The authors show that SCARIBOS has good performance for tidal water levels (at least at one location), but tides presumably only have a minor effect on particle dispersal around Curaçao compared to the prevailing currents and eddies (and reproducing water levels associated with tides does not necessarily mean tidal currents are reproduced well). The comparison in Figure 2 with ADCP data is very good, but this is one snapshot in time. There is some further assessment of the model in section 4.1, but this is vague and qualitative. The manuscript describes SCARIBOS as a whole (which covers a much broader domain than just Curaçao), but the hydrodynamics are only assessed immediately around Curaçao. I would make the following suggestions:

   a. Surface currents (mean state, and ideally variability) should be assessed for the *entire domain* against an (ideally) observational product, e.g. GLOBCURRENT (and this should be done quantitatively and/or graphically, not just described in the text). Remote-sensing surface current products for the region will have a much coarser resolution than SCARIBOS, and will of course not resolve interactions with the islands, but will at least allow the reader to be satisfied that SCARIBOS reasonably reproduces the large-scale current systems and their variability (I did this very quickly by eye and it looks like SCARIBOS does quite well).

      We appreciate the suggestion to assess the model's performance over the entire domain, despite the study's primary focus being mainly around Curaçao. Following your recommendation, we performed a graphical comparison between the GlobCurrent dataset and SCARIBOS for the entire domain, using monthly averages for the year 2022. We plotted the comparison in Figure 2. Accordingly, we added lines in 2.3 explaining the methodology (lines 169 to 172) and assessing the performance of the model (lines 176 to 190). Consequently, we changed the figure numbering of the entire new manuscript to match the changes.

   b. Given the importance of cyclonic eddies for the dispersal dynamics described in the manuscript, quantitative comparison of eddies simulated by the model to observations would also be useful, if possible, particularly since the ADCP observations were during a non-eddying time. The manuscript cites several papers that discuss eddy dynamics in the region, so I would ask the authors to consider whether there are observational data they could use for this assessment.

      Unfortunately, observational data in this area is scarce. The eddies we refer to that are reported in the previous studies are mostly observed in the middle of Caribbean Sea, north of the SCARIBOS domain. Moreover, none of the referenced studies (Carton and Chao, 1999; Murphy,

1999; Richardson, 2005; Van der Boog et al., 2019) covers the period of our simulation (2020-2024). For example, Carton and Chao (1999) focus on latitude around 14° for their analysis, which is beyond our model boundaries. Richardson (2005) analyses surface drifters, out of which very few trajectories reached our domain and none of them are associated with (anti)cyclonic eddies (as shown in fig. 11 of their paper). Likewise, Van der Boog et al. (2019) analyses pathways of anticyclones observed in mid-Caribbean Sea with none passing our domain either.

c.  The authors could also consider using Global Drifter Program drifter trajectories – if there are sufficient observations in this region – to compare true surface dispersal pathways (in a statistical sense) to simulated trajectories. This would be useful in the context of inter-island connectivity for Scenario 3.

That is a very good suggestion. However, unfortunately during our simulation period (2020–2024), there were very few Global Drifter Program trajectories passing through our model domain (see figure below).

[Figure]

*Figure: Drifter trajectories that passed our model domain (red box) in period 2020-2024. Data retrieved from Lumpkin and Centurioni (2019).*

Reference:

Lumpkin, R., and Centurioni, L.: Global Drifter Program quality-controlled 6-hour interpolated data from ocean surface drifting buoys, NOAA National Centers for Environmental Information, Dataset, https://doi.org/10.25921/7ntx-z961, [accessed: 17 January 2025], 2019.

2. I have a few concerns about the hotspot analysis (scenario 1):

Your suggestions and the suggestions from the Reviewer #2 below about improving/clarifying the HOSTPOT analysis are very insightful and useful. Therefore, we have carefully examined our analysis and plotting, and made (major) changes shown in Fig. 8 in the revised manuscript:

1. We re-run the HOTSPOT scenario in which we now released the particles every 12 hours instead of 24 hours. We did this to be consistent with the other two scenarios.
2. Based on Reviewer #2's suggestion, we revised the post-processing metrics to use the 'normalized unique particle count.' This metric tracks the number of unique particles that pass through each bin at least once during the simulation. By normalizing these counts, we ensure comparability across bins and provide a clearer representation of particle distribution patterns.
3. We calculated the time-averaged normalized unique particle counts across all simulated months. This approach highlights persistent patterns visible throughout most months, such as the shadowing effect along the southern coastline. More details below.
4. We re-wrote the methodology, results and discussion based on these findings:
   a. **Methodology:** lines 259 to 266, explaining how the normalized particle count is calculated
   b. **Results:** we re-wrote most of the section 3.2, due to changes in the findings derived from the new figure (Fig. 8). Main changes are because now we do not have PDF of particle concentration, and therefore the narrow band of the highest PDF in the northern coastline is not present anymore in the results. Previously this was the case because the particles are moving very slowly along this coastline (which resulted in very high PDF), but since we are now only counting the number of unique particles visiting each bin, this narrow band is not visible anymore. We now focus on the difference between the northern and southern coastline (Fig. 8a) and monthly variations (this part of the results did not change much). Changed lines 348-370 (most of section 3.2 is largely altered).
   c. **Discussion:** we changed from PDF to hotspots and from particle concentrations to particle counts in sections 4.2 and 4.3. The main conclusions stayed the same.

a. I'm finding it difficult to see the 'big picture' with this analysis, probably because there is so much variability in Figure 7 – perhaps the authors have tried this already, but I wonder if plotting a time-average (e.g. across the 'normal' state and 'eddy-dominated' states) might make spatial patterns clearer here.

To see the pattern that is observed across most of the months, we have created the time average of this normalized unique particle counts (averaging over all months, in revised manuscript Fig. 8a). We considered creating time-average across the 'normal' (i.e. 'NW flow') and 'eddy-dominated' state, but we found it too subjective and arbitrary to manually handpick the months associated with these two regimes, since it was often the case that half of the month was 'NW-flow-dominated' and the other half 'eddy-dominated'. Doing careful examination of these states would require much more detailed analysis into the exact distinguishing between the start and end of each state, which is out of the scope of this present manuscript.

b.  L47 defines hotspots as "areas where land-derived substances spend considerably more time than in other areas" – however, particles were seeded primarily in ocean grid cells, not coastal grid cells. If hotspots are genuinely supposed to reflect the fate of land-derived substances, particles should only be released from coastal cells (as was the case in Scenario 2). If the analysis is instead intended to identify zones of accumulation more broadly, the decision to release particles in a 1x1 degree square seems arbitrary, and I would be interested to know how sensitive the PDFs in Fig. 7 are to this decision (e.g. does the described accumulation of particles NW of Curaçao persist if particles are seeded over a broader area?). I also wonder if computing the average residence time of particles might be a useful way of identifying hotspots.

We agree with the comment that the original definition of hotspots does not fully align with the analysis presented in the manuscript. In response, we have revised our definition as follows (line 52): *In our study, hotspots are defined as areas where substances are more likely to accumulate than in other areas, potentially leading to increased stress on coral reefs.* This updated definition is also reflected in the Results section (section 3.2, line 347).

We would like to clarify that we are not specifically investigating land-derived substances in the hotspot scenario (this is addressed in connectivity Scenarios 2 and 3). Instead, this scenario aims to examine how well different areas around Curaçao are connected and to identify 'hotspot' areas visited by a higher number of particles, regardless of their origin. While we acknowledge that releasing particles in a 1x1-degree square is an arbitrary choice, we conducted a sensitivity analysis to assess the impact of reducing the number of particles released. Specifically, we examined the normalized unique particle counts when selecting only 1/2, 1/5, and 1/10 of the original particles (still within the 1x1-degree square). As shown in the figure below (with example month January 2021), the results are remarkably consistent across all cases. This demonstrates the robustness of our analysis, without needing to expand the release area as suggested. Interestingly, the pathways with higher normalized particle counts remain prominent, even with only 1/10 of the original particles selected. We added an interesting observation from our results in the discussion – linking our results with well-known Lagrangian Coherent Structures (LCS) in a new paragraph in section 4.2 (lines 466-471).

Finally, while we agree that computing residence times could provide valuable insights, it falls outside the scope of the current study. The spatial resolution of our model is too coarse to produce meaningful maps of residence times, particularly for coastal cells, which would likely be of greater interest for most applications.

[Figure]

*Figure:* Normalized unique particle count for January 2021 under varying particle release scenarios: (a) using the original number of particles, (b) using 1/2 of the original particles, (c) using 1/5 of the original particles, and (d) using 1/10 of the original particles.

c. The manuscript discusses hotspots in the context of the Island Mass Effect, evidenced by higher particle densities NW of Curaçao. This may be true, but I wonder if there is physical evidence from the model output (e.g. evidence of convergent surface currents in the lee of the island) that would further support this being due to the IME, as opposed to just being a consequence of NW-ward currents.

We have investigated the convergence of surface currents for two contrasting months. Our findings show convergence in this region, which aligns with the observed higher particle densities, supporting the presence of the Island Mass Effect (IME). Additionally, divergence along the southern coastline contributes to the observed shadowing effect. While we do not include this analysis in the current manuscript, it forms a key part of our ongoing research, which will be presented in the next paper.

3. I was interested in the Waitt Institute report that was cited in the manuscript, which appears to have various data (e.g. sewage indicators, trash accumulation indices, infrastructure density) that could be directly useful in this study – e.g. weighting coastal particles in Scenario 2 by infrastructure density to weight the connectivity matrix in fig. 8 since some parts of the coast are more likely to generate pollutants than others, or testing whether trash accumulation hotspots agree with predicted hotspot locations. I wonder if these data could be useful for model assessment?

Thank you for this suggestion. We agree that this has a lot of potential, however, in this manuscript we are not focusing on any specific pollutant or substance, and doing that would require much more elaborated and additional analysis on the sources and concentrations of these pollutants. Our experience indicates that pollutant and nutrient concentrations vary seasonally and are influenced by combination of factors like population density, geology, hydrology, and the presence of bays, all of which can create point sources of increased land-derived matter. The complexity of these factors is unfortunately beyond the scope of this present manuscript.

**Specific comments**

| Line | Comment |
| --- | --- |
| - | The maps in this manuscript (particularly Figure 1) appear to have been exported in a vector format, but this has made them very large (the size of the manuscript is 17MB) and causes lags when opening the file in a web browser. I would recommend exporting these maps in PNG format instead.
Thank you very much for this suggestion. Accordingly, we changed the format of the maps to PNG with 300 dpi. |
| 43 | This does not undermine the purpose of this study, but in the interest of accuracy/fairness I would consider citing the NCOM AmSeas model (https://www.ncei.noaa.gov/products/weather-climate-models/fnmoc-regional-navy-coastal-ocean), which covers Curaçao at 1/30 degree resolution.
Thank you very much for this suggestion. We have added the sentence in lines 44-47 and citation in lines 597-599. |
| 51 | Here and throughout (particularly section 2.3), I would use the word "assess" rather than "validate". Although most readers will know what you mean, the word "validate" in this context is arguably incorrect (e.g. Konikow & Bredehoeft, 1992).
We agree with your suggestion and have changed the wording accordingly throughout the manuscript. |
| 55 | Here and throughout, I would suggest being more specific about *which* substances the study is attempting to model (assumptions of positive buoyancy, no degradation, etc. will of course only be relevant for certain types of substances). This is particularly important in the context of section 4.3, as many of the pollutants that affect corals are not neutrally buoyant (e.g. sediment) and/or non-conserved (e.g. many nutrients).
Thank you for this suggestion. For better clarification we included non-degradable, positively buoyant substances in the second research question (line 59): |

| | |
|---|---|
| | *(2) How do these ocean currents affect the movement and distribution of non-degradable, positively buoyant substances at the ocean surface around the island, contributing to the formation of hotspots?*

Later in the manuscript we add *non-degradable* (line 248), in section 2.4 where we explain that our particles are: *passive particles, representing nutrients and pollutants… These particles, simulating **non-degradable**, positively buoyant substances, move with the surface flow conditions.*

We understand that substances such as nutrients and pollutants do not act this way, but we want to stress that we are making an 'idealized' case here. We explain this in section 4.3 (lines 488-491) as:

*Our study focuses on passive particles representing pollutants and nutrients, which are positively buoyant and non-degradable. These represent idealized conditions. In reality, many pollutants and nutrients behave beyond these assumptions, such as degradation and interactions with other substances, thereby highlighting the limitations and scope of our approach.* |
| 59 | Here and throughout, avoid the word "fine" in the context of resolution: it is subjective and context-dependent, just write the actual resolution.
We agree and we have changed in the manuscript accordingly. |
| Fig. 1 | This is a good figure, a couple of minor comments:
   - The contour increments in the colour map are slightly inconsistent with the tick labels in the colour bar (this is also the case for fig. 4).
   - The brown land polygons are slightly inconsistent (higher resolution?) with the grey coastline
Thank you very much for pointing these details – we have changed them accordingly. |
| 119 | I assume the vertical resolution is finer at the surface – if this is the case, I would specify it (and possibly give the range of values of the thickness of the upper layer).
Thank you for pointing this out. The vertical resolution is indeed finer at the surface, with the thickness of the upper layer ranging from a few centimetres to approximately 4 meters. We have included this information (lines 125-126). |
| 120 | Please justify how it was determined that 4 months was a sufficient spin-up duration.
A spin-up duration of 4 months was selected based on the assumption that the initial and boundary conditions provided by the GLORYS Copernicus model product enable the system to rapidly achieve a quasi-equilibrium state. The model uses a *'hot start'*, initializing with velocities (and salinity and temperature) interpolated from the GLORYS Copernicus model product. Additionally, we analysed the time series of the average eddy kinetic energy (EKE) across the entire domain and observed that the energy is stabilized quickly. The figure below shows the time series for the first 2 years, starting from the initial month of December 2019. This justification has been added to the manuscript on lines 127–129 and 145-146. |

[Figure]

*Figure:* Time series of Eddy Kinetic Energy (EKE) averaged over entire domain (black line), along with the EKE smoothed by a 12-hour moving average (red line). Time series spans from December 2019 to January 2022.

| 121 | How frequently were currents saved (hourly?), and were these snapshots or averages? What was the frequency of current data used in Parcels? |
| --- | --- |
| | We have added the additional information in lines 129-130: |
| | *Model outputs include hourly averages of horizontal and vertical velocities, temperature and salinity, stored for every grid cell in the domain ...* |
| | Additionally, in the description of particle tracking, we mention that Parcels uses the hourly average outputs from SCARIBOS for particle tracking (line 247). |
| 123 | I assume that smoothing was performed on the bathymetry (as I believe is standard for ROMS/CROCO preprocessing) since very steep slopes can cause stability issues. If this is the case, I think this should be mentioned in the methods. |
| | Indeed, that is the case. We have added this information in lines 136-137: |
| | *Smoothing of the bathymetry was performed using the CROCO TOOLS product (V1.3.1) to mitigate steep slopes that could cause instabilities in the model.* |
| 124 | The comment on "adjusting land grid cells" is very vague – I would briefly add the reason for these adjustments (presumably because the land-sea mask generated from the bathymetry is inconsistent with the true coastline). |
| | Indeed, that is the reasoning. We have clarified that in lines 138-141: |

| | |
|---|---|
| | *These adjustments are necessary to correct inconsistencies between the bathymetry-derived land-sea mask and the true coastline, ensuring more accurate representation of coastal features that significantly impact the formation and propagation of eddies around the islands.* |
| 134 | What is the source of the river discharge data (and is this based on monthly averages/monthly climatology…)?
 The source is a database by Dai and Trenberth, 2002, which we have referenced now in line 151:
 *… based on a climatological river discharge dataset by Dai and Trenberth (2002).*
 We have accordingly also added a new citation in lines 622-623. |
| 189 | Given the relatively limited role of tides in setting marine dispersal, and the fact that the ADCP comparison is based on one snapshot, I do not think we can conclude that "SCARIBOS accurately simulates surface-level dynamics, *making it a reliable tool for tracking surface currents*" on the basis of the evidence presented so far (see general point 1).
 Since we have added another assessment method following your advice (the GlobCurrent assessment), which specifically targets the surface currents across the entire domain, we decided to keep this statement here. |

| | |
|---|---|
| 204 | How did you determine the number of particles to be released? Was there a sensitivity analysis?
 We release them at the original spatial grid of the SCARIBOS model, because this is the finest resolution at which there is meaningful information in the data. |
| 204 | Is there a reason why particles were released every 24 hours for Scenario 1, but every 12 hours for Scenario 2? I don't think this is an issue, just seems a bit odd that different release frequencies were used.
 We have decided to re-calculate the hotspot scenario with release time of every 12 hours to have consistent release frequencies in all scenarios. |
| 256 | This mention of El Niño is a bit random – is it relevant to the local hydrodynamics that it's an El Niño year? This is again mentioned in line 368.
 We believe that it is possible that El Niño played a role in different hydrodynamics in year 2023 (there are notably more low-energy/eddy-dominated months in this year compared to other years), so we decided to mention it in the manuscript. |
| Fig. 5 | I really like these figures showing the monthly state of the model, I think they are very effective.
 Thank you very much! |
| 267 | Have there been hydrographic surveys in the region (or anything else) that supports this vertical velocity profile and the countercurrent?
 There is (very limited) data on the vertical extent from our expedition, but we decided not to include it here, as this manuscript (except from this paragraph) focuses on surface currents. The 3D circulation will be the topic of a follow-up manuscript. |
| 271 | I am not sure about tying this in with the AAIW… even if there is a signature of AAIW here, I would not have expected this to be relevant to the dynamics, which is the topic of this manuscript.
 We agree that this is not relevant, and since it is bold statement without much additional analysis, we decided to discard it from the manuscript. |

| | |
|---|---|
| Fig. 7 | I have a few comments about this figure:
Thank you very much for your careful inspection of this figure and for your useful suggestions!

- I would consider either (i) using a coarser binning grid, (ii) having more frequent particle releases, or (iii) outputting particle locations more frequently, because many of these sub-panels have a checkerboard pattern in the PDF. Using a coarser binning grid may also make the figure easier to interpret (by smoothing out some of the filamentation).
We have decided to increase particle releases (from 24h to 12h, as mentioned above) and to use more coarser binning grid. We made a binning grid of 100x100, matching the resolution of SCARIBOS. However, with all these adjustments we still did not avoid checkerboard pattern, as this is the nature of the results.

- Have you tried producing a time-mean (or seasonal, or non-eddy/eddy mean PDF)? Because there is so much variability, I am finding it difficult to see the 'big picture' from this figure.
As already mentioned above, we followed your advice and created time-mean of these results. We agree that this is a valuable contribution to the results, as it shows the general pattern without the reader needing to investigate each month separately.

- Zooming into the sub-panels, there is something going on in the background of Curaçao, see below (is the particle distribution being plotted on top of a raster outline of Curaçao)? I would remove this and keep the background fully white, otherwise this could be mistaken as representing data (high particle density).
Thank you very much for pointing this out. Indeed, this is a mistake in the plotting. We have repaired this in the revised manuscript (now Fig. 8).

[Figure]
 |

| | |
|---|---|
| Fig. 8 | Panel (a): because the changes in colour between some neighbouring areas are quite subtle, it may be helpful to readers to add numbered labels on the map next to each zone (this would also make it easier to quickly cross-reference between panel (a), and panels (b) and (c))
Panel (c): This is a really cool figure! I wonder if it might be useful to mark (e.g. with horizontal dashed lines) months associated with the passing of major eddies.
Thank you very much for these helpful suggestions! We added the numbering to the zones in panel (a) for easier cross-referencing in new manuscript, now Fig. 9.

Regarding your suggestion for panel (c), we decided not to mark the months associated with the passing of major eddies. This is because we did not want to manually handpick the months for the two regimes, as it was often the case that the months were split, with half of the month being 'NW-flow-dominated' and the other half being 'eddy-dominated' (as mentioned above). |
| 329 | However, most of the particles arriving at Zone 7 come from Zone 8 (and to a lesser extent, Zone 6), which are not highly populated. I understand that this study is not attempting to model any one particular pollutant (and therefore does not use a specific input function) but I do think it is important to qualify that, although Zone 7 may receive the most particles, this does not necessarily mean it will receive the greatest pollution burden. A similar point is made in the paragraph beginning on line 438.
Thank you for raising this point. While we understand your concerns, we intentionally avoid linking pollution impact solely to population density. The reason for that is that pollution sources on the island are influenced by various factors, something that other colleagues in the Sealink project focus on at the moment. For instance, complex hydrography, groundwater flow and illegal dumping of pollution in low-density areas. With our unbiased approach we give the chance for a reader to interpret the results based on the pollution source they are interested in. |
| 340 | Analyses for Scenario 2 assume that substances of interest have a lifespan of 30 days, which seems quite arbitrary. If you think this goes beyond the scope of this study then I accept that, but it would be interesting to know how sensitive Figure 8 is to particle lifespan.
We chose a conservative lifespan of 30 days, as our experiences show that most particles leave the area around Curaçao within just few days. |
| Fig. 9 | This could mess up the layout, but it might be useful to have a reference map (similar to fig. 8) to remind the reader of where these different sources are relative to Curaçao.
That is a very good suggestion. We have incorporated a reference map showing the particle locations after the first 50 hours for the example month of April 2020. This map is now included as Fig. 10a. |
| 382 | I am struggling to understand the point being made in this paragraph – can you relate this discussion of the IME to your results? Does the model predict large-scale downwelling in the lee of the island? Does the model predict upwelling (and divergence) along the south coast of Curaçao where there appears to be unidirectional offshore transport in surface waters?
Thank you for your thoughtful questions. While our model simulations do show upwelling, downwelling and divergence, a detailed analysis will be addressed in a follow-up study. In this manuscript, we focus primarily on the broader surface flow patterns, and we only mention the 3D circulation in a discussion as the preliminary observations. |
| 400 | I am not sure I understand why this paragraph is referring to the sticky water effect – my basic understanding (and as stated in the text) is that the 'sticky water' effect specifically refers to currents that are *retentive* but (as discussed in the previous paragraph), currents around the south coast of Curaçao instead appear to be highly dispersive – even though there are some similarities in the physical mechanism, i.e. current diversion around an obstacle. I understand the point being made, but is 'sticky water' the right term to describe it? Likewise in lines 433 and 467.
We agree that the term 'sticky' water seems counter intuitive. Our intention was to connect our observed phenomenon to previous studies and use a known term to provide context. While we acknowledge in our manuscript that the phenomenon we observe is quite the opposite, the key |

| | |
|---|---|
| | similarity is the lack of mixing between offshore and nearshore waters, which both phenomena share. Therefore, we prefer to keep the term 'sticky water', but now better explain (lines 449-450):
*Although the effect we observe differs fundamentally from the 'Sticky Water effect' (e.g., Andutta et al., 2012; Restrepo et al., 2014), both phenomena share a key similarity: the lack of mixing between offshore and nearshore waters.*

We changed the wording similarly in the next paragraph in lines 457-463:
*While the predominant northwestward Caribbean Current generally acts as a flushing mechanism, carrying particles away from the island and reinforcing the disconnection between nearshore and offshore currents, the reduced current speed during cyclonic eddies narrows the band along the southern coastline associated with this disconnection. ...[]... However, the hotspot analysis indicates that this nearshore disconnected band – similar in behaviour to the Sticky Water effect – although narrower, still persists during these events.* |
| 409 | Similar to the point on line 382, is there evidence (e.g. vertical velocities diagnosed from the model) that prove this is due to the IME? As mentioned in my comment on Fig. 7, I think this might be easier to see with a multi-year seasonal mean.
Thank you for your suggestion. We have decided to delete this paragraph, as the new hotspot post-processing analysis (normalized unique particle counts) no longer shows this phenomenon. Furthermore, as mentioned earlier, we are not analysing 3D patterns (such as vertical velocities) in the present manuscript and therefore do not have sufficient evidence to support these claims. |
| 419 | Change "reduce pollutant concentrations" to "reduce neutrally buoyant pollutant concentrations", since downwelling will *increase* the concentration of positively buoyant pollutants.
Thank you very much for this comment. We have changed it accordingly in line 479. |
| 442 | It might be useful to mark the location of major settlements on figure 4(b) and/or 8(a).
Thank you for your suggestion. However, we have decided to keep it as it is, as we have now added zone numbers next to the zones in Fig. 9a. This makes the figure already clearer and easier to interpret. |
| 444 | I am confused by the point being made here. What is meant by "the limited distance travelled by substances such as pollutants" – does that refer to their degradation timescale in the ocean? Surely that depends on the type of pollutant and, regardless, the results in this study still suggest that the south coast probably has low pollutant retention? Similarly, I don't understand the claim that there is "reduced dilution" (L448). Figure 7 shows that particles that start on the south coast quickly move away, and are not replaced by new particles. That sounds like dilution to me.
Thank you for pointing this out. To clarify, the statement about the limited dilution is based on the expectation that pollutants closer to the source are less dispersed due to limited travelling distance so far (compared to pollutants travelling large distances). While our study focuses on connectivity and not specifically on pollutant dilution (and neither degradation), we have rephrased the paragraph to make this point clearer. Revised manuscript lines 507-511:
*Although the retention times in these areas are relatively short, pollutants released near the source are likely to remain more concentrated locally because the particles do not spread as far before leaving this area, increasing the likelihood of localized impacts on coral reefs. While our study does not explicitly model pollutant dilution, it is reasonable to expect that areas closer to the source will experience higher concentrations compared to areas further away, where pollutants would disperse over larger areas.* |
| 453 | "Various substances" - I would suggest giving examples of substances carried by Venezuelan rivers that could be modelled by the approach taken in this study (i.e. positively buoyant, lifespan of longer than a month).
Thank you for this suggestion. We added some examples in lines 516-519: |

| | |
|---|---|
| | *… various substances, such as microplastics and microorganisms like bacteria and viruses attached to buoyant debris. Additionally, spill events from oil refineries along the western coast of Paraguaná, Venezuela, are among the documented sources of oil pollution in the region (Croquer et al., 2016).* With a new reference for Croquer et al. (2016) in lines 620-621. |
| 478 | Coral larvae are not always positively buoyant – their buoyancy declines with age (Szmant & Meadows, 2006). As they mature and gain vertical swimming ability, they have some control over their position in the water column, which varies considerably by species (e.g. Mulla et al., 2021; Tay et al., 2011). The point you are making here is fine, but I'd change "…which are also buoyant" to "…which often remain in the upper water column" (or similar). You are correct, thank you very much for pointing this out. We have changed it as suggested (lines 545-546). |
| 479 | Since there was no limit on the particle lifespan in Scenario 3, I think you need to quantify the connectivity timescale between Bonaire and Curaçao before you make this claim. I am not sure how coral disease is transmitted, but I assume the pathogens and/or vectors have a limited lifespan in the water column? We quantified the time scale of the fastest 5% of the particles which is roughly 40 hours. We added this information to line 551: *The timescale for such travel can be as fast as 40 hours when the currents are strong and in the right direction.* |
| 482 | I would restate in this sentence which specific areas you consider to be "critical areas" – at the moment, this is a very vague and generic statement. We agree and we have added a statement for clarification (line 558): *Identifying critical areas – such as regions with coral reefs that are particularly susceptible to high pollution loads – on a regional scale is essential…* |
| 493 | The configuration files should probably be archived on Zenodo or similar prior to publication, since GitHub is not a permanent repository. Thank you for your suggestion. We archived the configuration files at Zenodo, with DOI https://doi.org/10.5281/zenodo.14697794. We added this in reference to lines 569 and 604. |

**Technical comments**

| Line | Comment |
|---|---|
| 36 | Would rephrase to "Coral reefs are not just impacted by local sources of pollution, but also by broader environmental changes and anthropogenic activities" (otherwise it sounds like "broader environmental changes and anthropogenic activities" are an impact on coral reefs, rather than a source of impact). Very good advice – we have changed it accordingly (lines 37-38). |
| Fig. 6 | Change B) to (b) in the panel label. Changed. |
| 312 | Would change "western" to "westward" Changed. |
| 316 | This was mentioned earlier, but I would remind the reader at this point where the highly populated areas on Curaçao are. We added zones 3 and 4 (in brackets) for clarification (line 373). |
| 344 | Change 9A, 9B etc. to lower case. Changed. |
| 348 | Change "a strong signal" to "high particle transport" (or similar). |

| | | |
|---|---|---|
| | | Changed to 'high particle transport'. |
| 451 | | Consider changing "connection" to "upstream connection" |
| | | Changed. |
| 475 | | These sentences ("While this study focuses on… pollutants, its methodology can be adapted… for tracking plastic debris") makes it sound like plastic debris is not a pollutant. |
| | | Indeed, it may come across that way. In our manuscript we introduce our land-derived substances with examples of pollutants from the sewage system. Here we wanted to distinguish between pollutants that behave passively and resemble (dissolved) matter from the pollutants that have very different characteristics (such as size and shape → plastics). |
| | | We changed in the line 541 from _plastic debris_ to _buoyant (macro)plastic pollution too_. |

**References**

Konikow, L. F., & Bredehoeft, J. D. (1992). Ground-water models cannot be validated. _Advances in Water Resources_, _15_(1), 75–83. https://doi.org/10.1016/0309-1708(92)90033-X

Mulla, A. J., Takahashi, C. L. S., & Nozawa, Y. (2021). Photo-movement of coral larvae influences vertical positioning in the ocean. _Coral Reefs_. https://doi.org/10.1007/s00338-021-02141-7

Szmant, A. M., & Meadows, M. G. (2006). Developmental changes in coral larval buoyancy and vertical swimming behavior: Implications for dispersal and connectivity. _Proceedings of the 10th International Coral Reef Symposium_, _1_, 431–437.

Tay, Y. C., Guest, J. R., Chou, L. M., & Todd, P. A. (2011). Vertical distribution and settlement competencies in broadcast spawning coral larvae: Implications for dispersal models. _Journal of Experimental Marine Biology and Ecology_, _409_(1–2), 324–330. https://doi.org/10.1016/j.jembe.2011.09.013

---

## Author Comment (AC2)

**Summary**

Bertoncelj et al. introduce SCARIBOS, a regional configuration of the CROCO ocean model for the South CARIBbean Ocean System with a kilometre-scale horizontal resolution of 1/100 degree. The authors use four years (2020-2024) of surface velocity fields output by the model to undertake three Lagrangian particle tracking experiments using the OceanParcels framework to explore the surface connectivity of flows surrounding the island of Curaçao. The Lagrangian experiments allow the authors to investigate (a) potential hotspots of marine pollutants around Curaçao, (b) intra-island coastal connectivity, and (c) the connectivity between the Curaçao coast and the neighbouring Aruba, Bonaire, Venezuelan islands, and the Venezuelan mainland. The authors should be commended for the Figures, especially those relating to the Lagrangian analysis, which convey the central findings both clearly and creatively. The manuscript is generally well written, although a more comprehensive description of the numerical modelling approach (including model validation), further exploration of the role of mesoscale eddies in driving surface connectivity, and an improved discussion on the limitations and wider relevance of the findings is needed. I would recommend the manuscript for publication subject to major revisions addressing the comments made below and the excellent suggestions made by Reviewer #1.

We thank the reviewer for their detailed and very useful comments and suggestions. Below, we respond to each comment in detail and address them accordingly. Line numbers mentioned in the text correspond to line numbers in the revised manuscript.

**General Comments**

- ***The current title:*** 'Flow patterns, hotspots and connectivity of land-derived substances at the sea surface of Curaçao in the Southern Caribbean' feels like a description of what the author's set-out to investigate in this study but does not give any indication of the main findings or conclusions drawn from the analysis. Although the title is perfectly acceptable in its current form, I wonder whether it could improved to highlight the findings most relevant for stakeholders / policymakers.
  Many thanks for the suggestion. We have decided to stick with the current title.

- ***Model description and validation:*** Given that this is the first documentation of the SCARIBOS simulation and it is likely this will be used by other studies in the future, further details are needed to improve the model description, including highlighting the absence of Stokes Drift, stating whether a current feedback parameterisation is implemented, commenting on key parameterisation choices of ocean physics (e.g. in the surface mixed layer), and including the frequency at which velocity and tracer fields are output (this will be especially relevant for future studies). Moreover, as highlighted by Reviewer #1, the validation of the simulation is currently insufficient to robustly conclude that it accurately represents the circulation of the Caribbean Sea (see specific comments below).

Moving forward, I would suggest broadening the validation beyond Curaçao to the wider Caribbean Sea domain and to use several sources of observations, including surface drifters deployed in the region during the simulated period (2020-2024).

Thank you for pointing that out. We agree that more detailed description of the model is valuable for the future users, and we have added the information accordingly to lines 153-165. Consequently, we also added 4 new citations: Godfrey and Beljaars, 1991 (lines 646-647), Fairall et al., 2003 (lines 642-644), Jackett and McDougall, 1995 (lines 659-660) and Large et al., 1994 (lines 669-670).

Following the suggestion also by Reviewer #1 we have included additional validation by graphically comparing GlobCurrent (Copernicus product) and SCARIBOS for the entire domain for the monthly averages for year 2022. We plotted the comparison in Figure 2. Accordingly, we added lines in 2.3 explaining the methodology (lines 169 to 172) and assessing the performance of the model (lines 176 to 190). Consequently, we changed the figure numbering of the entire revised manuscript to match the changes.

Unfortunately, during our simulation period 2020-2024 almost no drifters from the Global Drifter Program were passing our model domain (see figure below).

[Figure]

*Figure: Drifter trajectories that passed our model domain (red box) in period 2020-2024. Data retrieved from Lumpkin and Centurioni (2019).*

Reference:

Lumpkin, R., and Centurioni, L.: Global Drifter Program quality-controlled 6-hour interpolated data from ocean surface drifting buoys, NOAA National Centers for Environmental Information, Dataset, https://doi.org/10.25921/7ntx-z961, [accessed: 17 January 2025], 2019.

- ***Extracting maximum value from the 1/100 degree SCARIBOS simulation***: the Lagrangian analyses presented in the study are all well motivated and provide new insights into surface connectivity in the region. However, on reaching the conclusions of the manuscript, I did not feel that the full potential of this state-of-the-science simulation had been extracted. Firstly, it would be valuable to know if the insights presented are critically dependent on resolving flow structures at 1/100 degree or could similar conclusions be drawn using a much lower resolution. One powerful way to illustrate this would be to spatially coarsen the surface velocity field and repeat one or more of the Scenarios to examine how this impacts marine connectivity. A natural caveat here would be that the coarsened velocity field still originates from one which explicitly represented (sub)mesoscale dynamics, however, identifying the critical threshold of horizontal resolution required to represent connectivity in this region would be an incredibly valuable contribution of this work and would extend beyond the present application to Curaçao.

  Thank you very much for this insightful comment and valuable suggestion. While conducting a sensitivity analysis with different spatial resolutions would indeed provide interesting and important insights, we believe this is beyond the scope of this manuscript. Our choice of a 1/100° resolution was specifically motivated by the need to resolve the shape of Curaçao and its surrounding features accurately. Previous models with coarser resolutions, such as 1/30° (model AmSeas, finest available at the moment), represented Curaçao poorly, with the island's shape incomplete and Klein Curaçao entirely absent (see figure below).

[Figure]

*Figure:* Model AmSeas with 1/30° spatial resolution (example screenshot). Figure found on https://www.ncei.noaa.gov/erddap/griddap/NCOM_amseas_latest2d.graph?surf_el%5B(2025-01-19T00:00:00Z)%5D%5B(10.066160202026367):(13.399160385131836)%5D%5B(291.16375732421875):(294.49676513671875)%5D&.draw=surface&.vars=longitude%7Clatitude%7Csurf_el&.colorBar=%7C%7C%7C%7C%7C&.bgColor=0xffccccff&.click=?32,134 (last accessible on 19 January 2025).

For the purposes of this study, accurately resolving the island's shape was essential, as the zones in Scenario 2 are relatively small, and the Lagrangian analysis of coastal connectivity requires precision.

Similarly, I think the authors could better exploit the eddy-resolving nature of the simulation to further investigate the role of mesoscale eddies beyond the primarily qualitative descriptions included currently. A valuable example of such an approach is Roach and Speer (2019) - https://doi.org/10.1029/2018JC014845 - which identified the timescales of variability in the flow field which are responsible for the connectivity between the Ross Gyre and the Antarctic Circumpolar Current by coarsening their 5-day mean velocity field to 90-day means and a single time-mean field. This allowed them to separate the connectivity associated with high-frequency (e.g., mesoscale eddies) and lowfrequency (e.g., seasonal variability) variability from the time-mean flow. Given SCARIBOS explicitly resolves the (sub)mesoscale, using such a time-coarsening approach in this study could provide new insights into the role of high-frequency flow features in establishing the surface connectivity around Curaçao. This may also o[er further retrospective justification for the use of a

1/100 simulation and underscore the value of such regional models for informing policy making and marine planning.

Thank you for a very interesting suggestion. Although we agree that this approach would definitely offer a new insight into the dynamics of the eddies, we believe that this is a very large additional step and is beyond the scope of the current manuscript.

**Specific Comments Abstract**

Lines 11-12: Suggest replacing ', as these substances...' with 'since these substances can be transported towards reef sites by ocean currents' given that its already implied that the substance has entered the ocean.

Corrected.

Lines 13-16: Suggest combining the two sentences beginning with 'SCARIBOS, a fineresolution...' and 'SCARIBOS covers the...' to reduce repetition. Then 'Furthermore,' can be dropped in the following sentence as it is not needed.

Corrected.

Lines 19-21: Suggest condensing these two sentences to be less ambiguous. For example: 'Our results reveal two dominant processes influencing the hotspot locations of positively buoyant substances...'

Thank you for pointing that out. We have changed the second sentence to (lines 20-21):

*These flow patterns influence hotspot locations, with higher accumulation of positively buoyant substances occurring during eddy events.*

Line 24: As a non-domain expert on marine pollutants, I was surprised to see justification of this work as providing valuable information for marine conservation and environmental management at the end of the Abstract. The need for kilometre-scale modelling of ocean connectivity to inform stakeholders' decision-making struck me as an important motivation for pursuing the study alongside the current 'problem statement' highlighting the general decline of coral reef communities.

Thank you for your comment. In this study, we aim to address a specific issue contributing to the decline of coral reef health—one that can be directly addressed through informed policy changes. While our modelling employs still a relatively large scale (1km resolution), our analysis demonstrates that the results have a potential to inform policymakers. For instance, in the coastal connectivity analysis, certain areas are identified as more likely to experience particle accumulation, highlighting zones of potential concern and intervention.

**Introduction**

Line 28: Suggest being more specific on the rates of coral reef decline, by how much has this changed already? Is this accelerating? This would further strengthen the motivation for the study.

Thank you for this suggestion. We added the information about the rate of decline, but unfortunately, there is no papers yet published on its acceleration. Here is the revised statement (lines 27-29):

*However, previous studies have reported a significant decline in coral cover, with more than 50% of living corals on shallow reefs lost between the 1970s and the early 2010s (e.g., Bak et al., 2005; Vermeij et al., 2011; Waitt Institute, 2017).*

Lines 30-32: Suggest combining these two sentences: 'are susceptible to accumulating pollutants, bacteria and viruses originating from urban areas, ...'

Corrected (lines 30-33).

Lines 34-35: As a non-domain expert, it would be interesting to know how the threat posed by the accumulation of marine pollutants and harmful biological substances around coral reefs compares to other threats, such as marine heat waves and ocean acidification. Why is this threat especially worth investigating?

Marine pollution is indeed one of the main threats to coral reefs, as highlighted in report Reefs at Risk Revisited by Burke et al. (2011). Unlike ocean acidification and thermal stress, which are global issues and more challenging to address directly, the accumulation of marine pollutants and harmful biological substances is a threat that can be mitigated relatively easily through targeted policies and improved local management. This makes it particularly worth investigating. However, we acknowledge that it is not the only threat to coral reefs.

Reference: Burke, L., Reytar, K., Spalding, M., Perry, A.: Reefs at Risk Revisited. Washington (USA): World Resources Institute. 130 pp. URL: https://www.wri.org/research/reefs-risk-revisited. 2011.

Lines 36-46: Suggest revising or restructuring this paragraph as it's currently difficult to follow: it begins with a recognition that remote sources of marine pollutants are important for coral reefs, then proceeds to discuss why existing numerical model simulations are insufficient to represent the local ocean dynamics of the Caribbean, and concludes that developing a SCARIBOS model is the answer to this challenge. I wondered whether an alternative framing of this paragraph could be to highlight that in almost all ocean general circulation models, most notably the CMIP6 ensemble, Caribbean small island states are (at best) represented by a single grid cell or not at all, hence we are currently not in a position to translate global nor regional scale insights to local communities (who may be significantly impacted by unresolved processes). Thus,

developing fine-resolution regional configurations like SCARIBOS provides a means to represent these regions more accurately and inform marine conservation and environmental management efforts.

Thank you for your thoughtful suggestion. While we understand your recommendation to reframe the paragraph, we would like to keep it as it is. We have structured the paragraph to first highlight the issue that pollution travels with ocean currents, thereby necessitating the use of regional circulation models. However, we also point out that no model with a fine enough resolution currently exists, and we are addressing this limitation with our work.

Lines 44-47: Suggest refining this conclusion to be more precise. The preceding text highlights that there is only limited high-resolution ocean model data for the Caribbean Sea, but why does it follow that we need a high-resolution model of Curaçao specifically. Is the coral reef environment here particular at risk or subject to emerging risks? Or are the ocean dynamics here representative of the wider region, such that the insights drawn from this study are applicable elsewhere?

Thank you for your feedback. While Curaçao is not uniquely at risk compared to other (neighbouring) islands, our study focuses on Curaçao as a case study due to the scope and objectives of our project. Although Curaçao is project-specific, we believe that our methodology can still be adopted to other regions (and SCARIBOS can potentially be used for other few islands in the domain too).

Line 49: Suggest modifying to 'our research investigates **the** dominant surface ocean current patterns and substance transport **pathways** around Curaçao…'

Thank you. We corrected the sentence based on your suggestion.

Line 50: Do you mean monthly to inter-annual **variability** rather than environmental changes? I found this to be ambiguous as ocean variability is discussed later in the text.

You are correct – it is variability. We have changed it accordingly.

Line 51: Suggest using vertical **extent** here and elsewhere rather than reach in this context.

Corrected.

Lines 59-67: Not entirely sure this paragraph summarising the methodology is necessary on reading the full text since the description of the model and methods directly follows the introduction. As a compromise, it could be condensed to focus on the development of SCARIBOS as an answer to the research question which is nicely presented in the previous paragraph.

Thank you for your suggestion. We agree and we have altered the paragraph to skip the unnecessary information about the model validation (lines 67-72).

**Methods**

Figure 1: Suggest here and elsewhere ensuring that the colourbar ticks and contour levels are aligned to make it clear that the sea floor in the regions shown in white, for example, lies between 0-500 m depth etc.

Corrected.

Line 92: More appropriate, given the geographical location of this study, to refer to the Atlantic Meridional Overturning Circulation since you are referring to the warm upper limb waters feeding the Florida Straits.

Agreed. We added 'Atlantic' (line 97).

Lines 120-121: Is this really a sufficient model spin-up time to ensure that the Lagrangian experiments are not capturing the ongoing adjustment of the mean flow and eddy kinetic energy fields? How was this determined? Further details should be included here.

A spin-up duration of 4 months was selected based on the assumption that the initial and boundary conditions provided by the GLORYS Copernicus model product enable the system to rapidly achieve a quasi-equilibrium state. The model uses a *'hot start'*, initializing with velocities (and salinity and temperature) interpolated from the GLORYS Copernicus model. Additionally, we analysed the time series of the average eddy kinetic energy (EKE) across the entire domain and observed that the energy is stabilized quickly. The figure below shows the time series for the first 2 years, starting from the initial month of December 2019. This justification has been added to the manuscript on lines 127–129 and 145-146.

[Figure]

*Figure:* Time series of Eddy Kinetic Energy (EKE) averaged over entire domain (black line), along with the EKE smoothed by a 12-hour moving average (red line). Time series spans from December 2019 to January 2022.

Lines 122-125: What is the spatial resolution of the bathymetry product linearly interpolated from? What manual adjustments were made to the bathymetry? Further details on how this was undertaken and to what extent this (presumably) improved the resulting flow structures around the island should be included.

Thank you for this comment. We added the spatial resolution of both bathymetry products and we further explained the smoothing of bathymetry and why we made manual adjustments (lines 136-141):

*Smoothing of the bathymetry was performed using the CROCO TOOLS product (V1.3.1) to mitigate steep slopes that could cause instabilities in the model. ...[]... These adjustments are necessary to correct inconsistencies between the bathymetry-derived land-sea mask and the true coastline, ensuring more accurate representation of coastal features that significantly impact the formation and propagation of eddies around the islands.*

Lines 129-131: Is the interpolation scheme used to downscale the GLORYS12V1 velocity and tracer fields bilinear or conservative (conservative-normed)?

The interpolation method used for GLORYS12V1 velocity and tracer fields is spline interpolation, which was a default option in CROCO TOOLS configuration.

Line 132: When applying the surface atmospheric forcing does SCARIBOS account for the current feedback to the atmosphere (CFB), which contributes to the oceanic circulation by damping mesoscale eddies. (i.e., Does CROCCO account for the fact that the ERA5 surface wind stress field acting on the ocean has already 'felt' the surface ocean currents and hence simulations forced without CFB overestimate the mean circulation and the mesoscale activity). An excellent discussion on the use of reanalysis winds to force ocean models in provided in Section 6 of Renault et al., (2020) - https://doi.org/10.1029/2019MS001715.

Current feedback to the atmosphere (CFB) is not taken into account in SCARIBOS. We acknowledge that this leads to overestimation of mean circulation and mesoscale activity. We have added this information to line 160.

Lines 134-135: At what frequency are the 2-D (surface) and 3-D velocity and tracer fields output from the SCARBIOS model? Are daily mean fields being used as the inputs to OceanParcels? This is currently unclear and should be added to both the model description and the description of the Lagrangian experiments undertaken in this study.

Velocity and tracer fields are outputted hourly and the same frequency (1h) is used as input in all Parcels simulations. We have added lines in the new manuscript accordingly:

Lines 129-130: *Model outputs include hourly averages of horizontal and vertical velocities, temperature and salinity, stored for every grid cell in the domain ...*

Lines 246-248: *Particle tracking simulations using the Parcels v3.0.3 framework (Delandmeter and Van Sebille, 2019) are conducted to model the movement of passive particles, representing nutrients and pollutants, using hourly velocity fields from the uppermost layer of the SCARIBOS model.*

Lines 189-193: Unfortunately, I do not think that sufficient validation has been undertaken to justify the conclusion that 'SCARIBOS accurately simulates surface-level dynamics'. In this section, the model has been shown to reproduce the sea level timeseries at a single location and time-average currents agree qualitatively with limited observations in magnitude and direction. Reviewer #1 has made a number of excellent suggestions on ways to improve this validation, which I will not repeat here. I strongly support the use of surface drifters to validate the surface flow field. More broadly, I also think that the authors should comment (either in the methods or discussion section) on the somewhat philosophical challenge of undertaking simulations at kilometre-scale resolution in regions where observations are sparse – how do we know what good looks like? This also relates to my more general comment; I think it would be valuable for the authors to consider what is the minimum horizontal resolution needed to investigate connectivity in the Caribbean Sea (see general comments above).

Thank you very much for these suggestions. As already stated above, we decided to make additional validation with the use of GlobCurrent Copernicus product (see Fig. 2 as the new figure in the manuscript and above for the details on new lines explaining the methodology and assessment).

Regarding the suggestion to comment on the challenges of conducting simulations at 1 km resolution in regions with sparse observations, we believe this is already understood by the community. The limitations of such models, particularly in under-observed areas, are common across similar studies. SCARIBOS represents the finest resolution currently available for this region, and like any model, it must be interpreted with caution. As such, we do not believe that further elaboration on this point is necessary.

Line 197: Suggest acknowledging here that the velocity fields do not include Stokes Drift rather than leaving this until the Discussion.

Agreed. In line 248 we have added the following sentence:

*These velocity fields do not include Stokes Drift.*

Lines 204-205: To be clear, does this equate to releasing a single particle in each 1/100 grid cell? How sensitive are the results of Scenario 1 to these initial conditions given that (sub)mesoscale turbulence is explicitly resolved in this simulation? Conceivably, given how chaotic the underlying velocity field is, a small difference in the initial position of a particle could result in a very different final position following 30 days of advection. An insightful discussion of the chaotic behaviour of Lagrangian trajectories is presented (albeit applied to ocean ventilation in a much coarser OGCM) in MacGilchrist et al. (2017) - https://doi.org/10.1002/2017JC012875.

You are correct, there is one particle per grid cell. Moreover, following the suggestion below we decided to re-run Scenario 1 in order to release the particles every 12 hours (to be consistent with the other two scenarios).

You are right to be concerned about the sensitivity of the results to the particle's initial position. To check this, we conducted a sensitivity analysis to assess the impact of reducing the number of particles released. Specifically, we examined the normalized unique particle counts (see below for more information) when selecting only 1/2, 1/5, and 1/10 of the original particles (still within the 1x1-degree square). As shown in the figure below (with example month January 2021), the results are remarkably consistent across all cases. This demonstrates the robustness of our analysis. Interestingly, the pathways with higher normalized particle counts remain prominent, even with only 1/10 of the original particles. We added an interesting observation from our results in the discussion – linking our results with well-known Lagrangian Coherent (LCS) in a new paragraph in section 4.2 (lines 466-471).

[Figure]

*Figure:* *Normalized unique particle count for January 2021 under varying particle release scenarios: (a) using the original number of particles, (b) using 1/2 of the original particles, (c) using 1/5 of the original particles, and (d) using 1/10 of the original particles.*

Lines 205-206: The statement: 'The internal particle simulation timestep is set to 5 min and trajectories are archived every hour' is repeated for all three experiments. Suggest outlining the common features of the three scenarios in a final paragraph to reduce repetition.

Thank you for your suggestion. We made changes accordingly.

Lines 209-210: Did you also consider calculating Lagrangian PDFs by counting the number of unique entries into each given grid cell normalised by the total number of particles (i.e, calculating the likelihood that any given particle will enter a grid cell at least once during its lifetime)? In my experience, this can improve the clarity of Lagrangian PDF plots where recirculation features are dominant and better illustrate the net flow pathways.

Thank you for this very insightful suggestion. We decided to follow your advice and instead of PDFs we calculated the unique particle counts normalized by the total number of particles. We re-wrote the methodology, results and discussion based on these findings:

**Methodology:** changed lines 259 to 266, explaining how the normalized particle count was calculated.

**Results:** we re-wrote most of the section 3.2, due to changes in the findings derived from the new figure (Fig. 8).

Main changes are because now we do not have PDF of particle concentration, and therefore the narrow band of the highest PDF in the northern coastline is not present anymore in the results. Previously this was the case because the particles are moving very slowly along this coastline (which resulted in very high PDF), but since we are now only counting the number of unique particles visiting each bin, this narrow stripe is not visible anymore. We now focus on the difference between the northern and southern coastline (Fig. 8a) and monthly variations (this part did not change much). Changed lines 347-370 (most of section 3.2 is largely altered).

**Discussion:** we changed from PDF to hotspots and from particle concentrations to particle counts in sections 4.2 and 4.3. Main conclusions stayed the same.

Lines 215-216: Why are particles released every 12 hours in Scenario 2 compared with every 24 hours in Scenario 1? Is the initial time of release important relative to the diurnal cycle of atmospheric forcing, what time each day does this take place? It would be helpful for the authors to comment more on these uncertainties.

As mentioned earlier, we adjusted the release interval to 12 hours for Scenario 2 to maintain consistency across all scenarios. While atmospheric forcing might introduce some variability in our region, using a 12-hour interval helps to mitigate any potential bias related to diurnal atmospheric cycles. Moreover, in this region tidal forces are relatively weak, so we believe the precise time of release does not significantly influence the results. If tidal forces were stronger and particles were released at specific tidal phases, this could introduce a bias in the initial deflection of particles.

Lines 225-226: Why are particle released two coastal grid cells away? Presumably, beaching of particles could be a problem when using a numerical time-stepping scheme to determine the trajectories?

Particles are released within the first and second coastal grid cells away from land, forming a ribbon-like area encompassing these two grid cells. We chose to release particles in two grid cells rather than one to better represent the wider area from which they originate. In the OceanParcels code, particles driven by the hydrodynamic model, which is based on a C-grid (in our case CROCO is based on a C-grid), are not expected

to beach (i.e., they do not get stuck on land grid cells). Therefore, this issue does not affect our results.

Lines 236-237: Suggest adding a brief description of the locations of the remaining 1% of particles in Scenario 3 which do not leave the domain. Are these locations consistent between particle releases? If so, would regions of high particle persistence be particularly concerning for marine pollution and environmental management?

We appreciate your suggestion and have added the requested information in the results section 3.4, lines 415-418:
*Moreover, since our simulations were designed to terminate early if the computations take too long and only 1% of particles remained in the domain, this primarily occurred in the simulations of the Venezuelan mainland, due to the large number of particles released and the low currents near the mainland. Most of the remaining particles are located near the Venezuelan mainland, particularly along the east coast of the Paraguaná Peninsula.*

Figure 4: Excellent Figure, the authors have done a great job of visualising the differences between the Scenarios. Suggested modification to the final line of the Figure caption: 'The destination area highlighted around Curaçao represents the region within which particles are tagged as reaching the Curaçao coast.'

Thank you very much for the compliment and for your suggestion! We have changed the caption accordingly (lines 295-296).

**Results**

Lines 250-257: In Figure 5b, interannual variability appears to dominate over seasonality, so I would suggest caution not to overinterpret monthly behaviour based on four years of surface velocity data. Caveating the discussion by highlighting the limited number of months available to sample (4 instances each) would be one approach.

We added 'inter-annual' variability in this paragraph (line 305) to avoid being too biased with only mentioning 'seasonal'. We do acknowledge the limiting 4 years of simulations at the very beginning of the discussion (section 4.1, lines 429-430):

*While definitive conclusions are limited by the four-year simulation period, the observed seasonal to inter-annual patterns, even during El Niño year (2023), provide strong evidence for the robustness of these processes.*

Lines 263-266: Why is the analysis restricted to a single meridional cross section? A comparison of the 2-dimensional (longitude-latitude) flow field at various depths would properly account for the spatial dependence of the flow and to make the conclusion that the surface velocity field is representative of the upper ocean flow field more robust.

We chose to focus our analysis on a single meridional cross-section because this specific cross-section provides significant insight into the dynamics of the system. Our analysis concentrates on the zonal flow at this passage, as surface currents clearly indicate that the zonal flow is strong in the region between Venezuela and Curaçao (and definitely much stronger than the meridional flow). While we agree that analysing multiple cross-sections or a 2-dimensional flow field would provide a more detailed spatial perspective, we believe that the data from this single cross-section adequately captures the thickness and behaviour of the surface flow in this region.

Lines 270-272: Are the T-S properties of the westward current consistent with AAIW at these latitudes? This would strongly support the inference.

Following the comment from Reviewer #2 we decided remove mentioning of the AAIW in our manuscript.

Lines 273-280: This paragraph is quite confusing. The opening sentences largely repeat the findings above and third sentence seems to preface the say that the flow field is highly variable. I would also recommend removing 'observed' on Line 277 since (I think) you are still referring to the output of the SCARIBOS model here?

Thank you for pointing this out. We re-wrote this paragraph as (lines 327-331):

*Monthly variations of the zonal velocity along the meridional cross-section (Fig. 7b) reveal a dynamic two- to three-layer system, with a distinct westward surface current often overlaying a deeper counter current, as shown in the average profile (Fig. 7a). However, this system sometimes undergoes significant changes under specific scenarios. Firstly, during periods when the northwestward directed surface current dominates, it can extend to greater depths than average, leading to the absence of a distinct undercurrent...*

Lines 289-290: This concluding sentence feels slightly disjointed from the preceding text, which is a very nice synthesis. Perhaps, the component of the discussion that is missing is: is it reasonable to assume that the vertical motion of marine substances limited to the upper 10-20m which the surface velocity field is representative of?

We understand your concerns and we address this matter in the discussion section 4.2, specifically in the last paragraph (lines 472-474), where we acknowledge that we are only considering surface currents and not the full vertical transport dynamics. With slight alternations to the text:

*Finally, it is important to note that the particle simulation only considers the surface ocean layer, which we correspond to the top 10-20 m of the water column. While this approach provides a reasonable approximation of average conditions within this depth range, it neglects vertical movements and transport, which are crucial for studying nutrient fluxes. ...*

Line 295: Suggest adding 'reveals significant monthly and inter-annual variability'. More generally, it would be interesting to assess statistically whether the variability seen between monthly release maps is stochastic versus seasonal-interannual in nature. A similarity metric, such as the Fraction of Unexplained Variance (FUV) could be used to compare months and assess how similar any given month is to its monthly climatology (e.g., what fraction of the PDF shown for April 2023 can be explained by the April (2020-2023) average).

Thank you for your suggestions. We have updated this section of the results, incorporating the term 'inter-annual' in line 357 to better reflect the variability observed.

While we agree that computing metrics such as the Fraction of Unexplained Variance (FUV) could provide additional insights, we believe this analysis extends beyond the scope of our study. The comparison between the PDF (or, in our revised analysis, the unique particle count) and parameters such as current speed or direction is complex. The PDF is influenced by a wide range of factors, including speed, direction, convergence, and eddy kinetic energy (EKE), making the interpretation of such metrics non-trivial. Conducting this type of statistical comparison would require a more extensive investigation that lies outside the primary objectives of this manuscript.

Figure 8: As a non-domain expert, I found the large number of connectivity matrices in (c) to be difficult to interpret and, in contrast to the other Figures presented in the manuscript, to be the least effective at highlighting the key result of this Scenario. Two possible suggestions, which the authors are fully entitled to disregard, would be to replace the Source Zone numbers with geographical names as in (a) colouring font according to their location, and either masking or recolouring the 100% connectivity boxes (this value is known since particles are released here, but is the boldest feature in every subplot).

Thank you for your feedback. We have made several adjustments to improve the readability of the connectivity matrices (in revised manuscript Figure 9). We now placed the zone numbers next to the corresponding zones in panel (a) to make it easier to interpret the results. Additionally, we have changed the display of 100% connectivity values by creating black crosses over the white background.

**Discussion and Conclusions**

Lines 370-372: This sentence could be clearer, suggest modifying to: 'There is {broad/strong/good} agreement between the surface current vectors simulated by SCARIBOS and those estimated from Lagrangian surface drifters in the Caribbean Sea between 1989 and 2003 (Richardson, 2005).'

Thank you for the suggestion. We have incorporated the proposed modification into the manuscript (lines 185-186). We also moved this paragraph to section 2.3, following your suggestion below.

Lines 369-375: Much of this discussion would be better placed in the methodology section validating the SCARBIOS model. This would ensure readers have greater confidence in the simulation's ability to represent the circulation in the Caribbean Sea before it is applied in the Lagrangian analysis, rather than discussing this retrospectively.

Thank you for the suggestion. We moved this paragraph to lines 184-190.

Lines 376-381: This brief discussion on cyclonic eddies is interesting, but I feel more could have been done in the Results to explore this (see earlier general comments on temporal coarsening), including coarsening the flow field in time to extract the signature of high-frequency flow components on the connectivity and persistence of marine pollution around Curaçao.

We agree this would be an insightful addition, but we believe it goes beyond the scope of our study.

Lines 435-437: This raises an interesting discussion point on the residence times of particles around the coast, however, residence times were not addressed in the Results section. Was this intentional and based on a supplementary analysis of the particle residence times? It would be interesting (perhaps in future work) to combine the findings on the connectivity of positively buoyant marine substances with their residence timescales in coastal regions around Curaçao, since a highly connected reef with a low flushing (high residence) time scale would surely be more susceptible to marine pollutants.

Thank you for your insightful comment. Indeed, exploring residence times is an important topic for future work. We also understand that residence times should be properly accounted for, and we believe this can be achieved more accurately with 3D simulations.

Line 459-460: The absence of wave effects, including Stokes Drift, should be commented on in the methods section as it's an important limitation of the surface velocity field used in the Lagrangian analysis.

Thank you – we added Stokes drift in the methods section (line 152).

Lines 474-484: This concluding section on 'Implications and future directions' could be improved by emphasising the value of the SCARIBOS 1/100 simulation – what have we learnt with this model which is not attainable at lower resolution – and identifying several future questions which will directly inform policymakers and stakeholders. Currently, plastic debris, coral larvae and marine pollution are discussed collectively, but it would be interesting to know how SCARIBOS outputs could be used in each case, thereby underscoring its long-term value as both a scientific and societal resource. For example, could SCARIBOS be used to predict (or train an ML model to predict) pollutant spills and the resulting environmental impacts?

Thank you for your suggestion. We have revised this section to place more emphasis on the value of the 1/100° resolution (lines 542-545):

*By taking the advantage of 1/100° resolution SCARIBOS simulation, it is possible to resolve smaller-scale transport pathways and retention zones that would remain unresolved in coarser models. This provides critical insights for predicting how plastic debris accumulates in specific areas, which can directly inform mitigation strategies.*

We have also added more detail on how SCARIBOS can be applied to other cases, such as coral larvae dispersal (lines 547-548) and oil spills (paragraph in lines 552-556).

---

## Referee Report (RR1)

**Summary**

I gratefully thank the authors for having taken my comments into account and for their efforts to revise the manuscript. This revised version reads well and resolves almost all of my original concerns on the model's limitations and wider relevance of the study's findings. I have just one general comment concerning the updates to the model validation and several minor comments motivated by the authors' revisions. I would recommend the manuscript for publication subject to any minor revisions needed to address the comments made below.

**General Comment**

While I am very grateful to the authors for responding to mine and Reviewer #1's concerns regarding the rigorous assessment of the SCARIBOS model, I still believe that there is an opportunity to improve the validation of the simulation included in the revised manuscript.

The authors' addition of a graphical comparison between the GlobCurrent dataset and SCARIBOS model is a welcomed one. However, I would strongly recommend regridding the SCARIBOS model output onto the coarser grid of the GlobCurrent data to allow for an equivalent comparison between the two products. This would also enable the authors to present the [model - obs.] bias of the simulated surface current field when compared with GlobCurrent observations. Naturally, there would still be a caveat that the SCARIBOS model includes more than just the Ekman and geostrophic contributions to the surface velocity field, but this would still be more informative than the current visual comparison.

On a similar note, I believe that Figure 4 could be improved by visualising only the model surface current velocity vectors co-located with the ADCP observations while retaining the background colour contours representing the surface currents of the entire model domain. This would allow for more of an 'apples-to-apples' comparison since the authors have already selected data for the observational period between 4th and 22nd January 2024.

Finally, I believe greater attention should be paid to validating the SCARIBOS model's ability to represent ocean properties, given that communicating the model's overall fidelity will assist future researchers considering using the simulation in their own work. In particular, I would suggest validating the sea surface temperature and sea surface salinity fields outputs by the SCARIBOS model against relevant observations (e.g., OISSTv2 – see Huang et al., 2021 - and the Multi Observation Global Ocean Sea Surface Salinity product – see Droghei et al., 2016). Moreover, it also occurred to me that the GO-SHIP / CLIVAR Repeat Section A22 (last completed in 2021) intersects the SCARIBOS model domain, and thus may provide an insightful meridional-cross section

with which to compare the model's temperature and salinity field as part of an improved Figure 7 (which does not currently include observations).

I would like to emphasise that the suggestions above are intended to give the scientific community even greater confidence in the SCARIBOS model, and hence encourage the wider use of this simulation beyond this particular study (which itself will serve as the documentation and validation of the model going forward).

**Minor Comments**

**Line 298-299**: When does particle seeding conclude in Scenario 3? I could not determine from the current text whether particles are still being released during the final month of the simulation (i.e., February 2024) and, if so, how are these dealt with in the Lagrangian analysis? For example, do you take into consideration the much longer advection time for those particles released in 2020 compared with those released in 2024?

**Figure 8**: Suggest using two separate colorbars for the upper and lower plots given that the colorbar is saturated in the lower plot, but not in the upper plot. This makes the upper plot appear washed out and does not highlight any prominent pathways.

**Lines 391-394:** Could you quantify this description of strong flushing events somehow? For example, is it the case that particles released in DJFM are more likely (higher proportion) to leave the model domain via the northern and western boundaries compared with those released throughout the rest of the year?

**Figure 9:** What is special about 2021 for the connectivity of Source regions 6, 7 & 8? This interesting anomalous lack of connectivity feels worthy of comment; what is this related to in the surface circulation?

**Section 3.4:** One intriguing question that came to mind re-reading this Section was: what conditions give rise to the anomalously high and low coastal connectivity values shown between Bonaire and the Venezuelan Islands and Curaçao. We'd certainly care strongly about instances of the former, especially in the context of marine pollution and its wider impacts. Are these episodes of enhanced coastal connectivity predictable?

---

## Author Response (AR2)

Dear Editor and Reviewers,

We sincerely appreciate your thorough review and constructive feedback on our revised manuscript, "*Flow patterns, hotspots and connectivity of land-derived substances at the sea surface of Curaçao in the Southern Caribbean*," submitted to the *Ocean Science*.

We have carefully addressed all the feedback provided and implemented the suggested revisions in the manuscript, with particular attention to the additional validation of the model. Along with the revised version, we are submitting a detailed response letter addressing your comments and suggestions, as well as supplementary material consisting of six figures: Fig. S1 representing the EKE time series and Figs. S2-S6 providing additional model validation analysis.

We hope that these revisions adequately address all concerns and further enhance the manuscript. We look forward to your feedback and hope that the manuscript will be suitable for publication in *Ocean Science*.

We appreciate your time and consideration.

Sincerely,
Vesna Bertoncelj (on behalf of all co-authors)

**Response to RC1:**

We greatly thank the reviewer for their valuable comments and suggestions on our revised manuscript. Below, we respond to each comment and address them accordingly. Line numbers mentioned in the text correspond to line numbers in the revised manuscript.

L65: I'd suggest changing "community model" to "ocean model" or "regional ocean model".

Agreed. We changed it to "regional ocean model".

L123: The precision for the grid dimensions (5 significant figures) feels excessive – the exact dimensions are in the model configuration file anyway. I would consider just writing 1.1 km.

Agreed. We changed it to 1.1 km.

L127: The EKE plot in the reviewers' response was useful, and I think this is an important justification for the spin-up duration of 4 months. This figure (integrated EKE vs time) would ideally be included in the supplementary materials but, otherwise, I would at least mention this in the main text (i.e. that the 4 month duration allowed EKE to stabilise across the model domain).

We thank the reviewer for this suggestion. We added the sentence in lines 129-130:

> This duration allowed that the eddy kinetic energy (EKE) stabilized across the model domain (Fig. S1), ensuring the system reached a steady state before the analysis period began.

Additionally, we added the figure showing the EKE time series as a Supplimenary Figure S1.

L178-180: I don't think this explanation for the difference between SCARIBOS and GlobCurrent is correct. GlobCurrent is based on observed SSH, so the effects of islands are included (since they exist in reality and therefore affect the observations) even if they are not resolved in the gridded GlobCurrent product. GlobCurrent can only capture the geostrophic and Ekman components of the surface flow, so it would not capture any ageostrophic effects of the islands on the flow. However, the differences between SCARIBOS and GlobCurrent are not just immediately around/downstream of the islands. SCARIBOS does not include data assimilation, so some differences will just be due to turbulence. There are some consistent differences though, e.g. most of the westward surface transport in GlobCurrent is north of the islands, whereas SCARIBOS simulates strong flow south of the islands as well. These differences are notable as they could affect the results of the manuscript.

We thank the reviewer for these insightful comments and for pointing out the issue in our explanation. We have revised the paragraph to clarify the differences in strong flow positioning between SCARIBOS and GlobCurrent and the possible reasons for these differences. Moreover, we compared SCARIBOS with GLORYS12V1 model (the model used for initial and boundary conditions) and found strong agreement between these two models (we added this comparison to Supplementary Fig. S2). This suggests that the differences from observational dataset by GlobCurrent likely arise from the choice of the boundary conditions. The changes are in lines 185–193:

> SCARIBOS effectively captures months with stronger currents, particularly in February, March, April, July and December 2022, where differences arise from the positioning of the core of the current. Notably, SCARIBOS simulates strong flow both south and north of the islands, while GlobCurrent shows stronger currents primarily north of the islands. These differences may stem from SCARIBOS resolving finer-scale flow interactions around the islands, including ageostrophic effects, whereas GlobCurrent, while incorporating the large-scale influence of islands through sea surface height observations, does not resolve their small-scale effects explicitly. When comparing SCARIBOS with GLORYS12V1 (Fig. S2), a much stronger agreement is observed, with both models showing strong currents both north and south of the island. This is expected, as GLORYS12V1 provides the boundary conditions for SCARIBOS, indicating that the differences with GlobCurrent likely arise from the choice of boundary conditions.

L260: "when particle" -> "when a particle", and "a grid" -> "a grid cell"

We have changed the text accordingly.

L310: I'm still not convinced by this mention of El Niño. The authors wrote in their response that they "believe it is possible that El Niño played a role in the different hydrodynamics in 2023 [because] there are notably more low-energy/eddy-dominated months", but a 4-year simulation is nowhere near enough time to confidently attribute something to El Niño (as opposed to some other mode of variability or random chance). If there are other studies that investigate the impacts of El Niño on eddy activity in the region then please cite them and explain the connection, otherwise I do not think it is appropriate to mention El Niño.

We thank the reviewer for pointing this out. We decided to follow the reviewer's advise and deleted the statement (line 350). Furthermore, we deleted the mentioning of El Niño in line 475.

L350: "origin" -> "origins" or "source regions", and "Contrary" -> "Conversely" or "On the contrary"

We have changed these accordingly.

Fig. 9: I preferred the old version of the figure where source=destination cells were filled, although I understand why the authors chose to block them out. However, I would at least suggest exporting sub-panels 9(b) and (c) as vectors, because the rasterised crosses don't look great at low resolution.

We decided to cover the cells with crosses following suggestions from Reviewer #2, as we agree that in this way the most notable highlights in the figure are not the 100% self-connectivity cells. We follow the reviewer's suggestion and have exported the figure now as vector for better resolution.

L438: I'm struggling to see how the results of this study are being attributed to the Island Mass Effect. In the original figures, it looked like there was particle accumulation immediately downstream of the island, but this is no longer the case in the revised figure 8. There's certainly perturbation of the flow, but the IME is more than just perturbation of the flow – it specifically relates to vertical movement. The authors wrote that they have further evidence for this which will be part of a future manuscript, but I am not reviewing that manuscript, and I can't see convincing evidence for the IME in this manuscript.

We follow the reviewer's suggestion and described observations without mentioning Island Mass Effect. The changes are in lines 482-486:

> *The interaction between currents and the island perturbs the strong northwest-directed oceanic flow. This current-island interaction results in generation of eddies in the lee. The strong horizontal divergence leads to significant differences in speed between currents upstream and in the lee of the island. In our study, this effect is particularly notable during periods when the currents are strongly northwest directed. In these periods, vortices form due to the island's influence on flow dynamics, resulting in reduced flow strength in the northwest of Curaçao.*

L449-452: Similarly, I am still unconvinced about this comparison with the Sticky Water Effect. The authors have now clarified that the effect observed in this study is not the Sticky Water Effect… in which case, why mention it at all? Just describe the observations.

We follow the reviewer's suggestion and described observations without mentioning Sticky Water Effect. The changes are in paragraph in lines 488-493:

> *Understanding the movement of particles is crucial for evaluating how substances (e.g. pollutants and nutrients) are transported around Curaçao. The monthly simulations of hotspots around Curaçao reveal a complex dynamic, particularly along the southern coastline, where a notable disconnection exists between the nearshore currents and those passing further offshore. Particles released nearshore are primarily transported offshore by surface currents. Concurrently, particles released from locations further away rarely reach the coast due to the separation between the nearshore flow and the ambient offshore currents that pass at a distance from the island. This results in a low accumulation of particles near the southern coastline.*

L466-471: I would remove this discussion of Lagrangian Coherent Structures from the manuscript. I can intuitively see why regions of higher normalized unique particle count might correspond with attracting LCS, but is this a physically rigorous way of identifying LCS? I am not familiar enough with LCS theory to answer that, but I suspect it is not. This would be an interesting future study, but this discussion feels out of place in the present manuscript given the lack of evidence provided. I certainly think the claim "This highlights the critical role that LCS play in shaping surface substance transport around Curaçao" should be removed, as should the Gomez-Navarro et al. 2024 reference (it does not use the same region or same method, so don't see how it is relevant apart from the fact that it deals with LCS). The Haller (2015) reference is also missing from the reference list.

We agree that this is not enough to make such claims and would be an interesting further study. We decided to follow the reviewer's suggestion and delete the entire paragraph on LCS from the manuscript including the references associated with it.

L509: I cannot see any evidence for the claim that "particles do not spread as far before leaving the area" – no particle dispersion metrics (e.g. Lyapunov exponents) were computed in this study.

We agree with the reviewer's observation and have revised this paragraph (lines 538-542) to focus on the fact that Willemstad, as a city, presents a much higher risk of spreading pollution compared to uninhabited zones:

> *Furthermore, the connectivity analysis indicates that the areas adjacent to the capital city Willemstad (zone 4, Fig. 9a) are also largely exposed. Urban runoff from this region poses a significant threat to nearby coral reef communities, as pollutants released here are likely to have greater impact on coral reefs. While our study does not explicitly model pollutant sources and concentrations, it is reasonable to expect that areas closer to Willemstad face higher risks compared to more remote areas, such as Klein Curaçao, which is uninhabited and unlikely to contribute significant pollution.*

L553-554: The mention of machine learning is a bit random. If the authors really want to include this then OK, but particle tracking is not particularly expensive in the first place, so I don't see why this application is specifically notable.

We thank the reviewer for this feedback. We have removed the mention of machine learning and re-wrote this paragraph to focus more on operational application of the methodology. Revised text in lines 583-586:

> *SCARIBOS simulations and the Lagrangian connectivity pathways could also be applied to predict pollutant spills and their potential environmental impact. By adapting our methodology, the model could be expanded for operational use by incorporating forecasted boundary and forcing conditions. This would enable more accurate predictions of pollutant transport and support the development of rapid-response tools for environmental risk management.*

Data availability: Is there a plan for how readers can access the full dataset if the two contacts were to leave their institution and no longer had access to these email addresses? Is there not a more sustainable solution for making this dataset available?

The model outputs will be available through https://dataverse.nioz.nl/, additionally we provided the general Research Data Management mail.

**Response to RC2:**

**Summary**

I gratefully thank the authors for having taken my comments into account and for their efforts to revise the manuscript. This revised version reads well and resolves almost all of my original concerns on the model's limitations and wider relevance of the study's findings. I have just one general comment concerning the updates to the model validation and several minor comments motivated by the authors' revisions. I would recommend the manuscript for publication subject to any minor revisions needed to address the comments made below.

> We greatly thank the reviewer for their valuable comments and suggestions on our revised manuscript. Below, we respond to each comment and address them accordingly. Line numbers mentioned in the text correspond to line numbers in the revised manuscript.

**General Comment**

While I am very grateful to the authors for responding to mine and Reviewer #1's concerns regarding the rigorous assessment of the SCARIBOS model, I still believe that there is an opportunity to improve the validation of the simulation included in the revised manuscript.

The authors' addition of a graphical comparison between the GlobCurrent dataset and SCARIBOS model is a welcomed one. However, I would strongly recommend regridding the SCARIBOS model output onto the coarser grid of the GlobCurrent data to allow for an equivalent comparison between the two products. This would also enable the authors to present the [model - obs.] bias of the simulated surface current field when compared with GlobCurrent observations. Naturally, there would still be a caveat that the SCARIBOS model includes more than just the Ekman and geostrophic contributions to the surface velocity field, but this would still be more informative than the current visual comparison.

> We thank the reviewer very much for this suggestion – we agree that regridding SCARIBOS improves the comparison, and we have changed it accordingly (Figure 2). Moreover, all the supplementary figures that we have added to the manuscript (as stated below) also include regridded SCARIBOS outputs to match the spatial resolution of the compared model/observation.

On a similar note, I believe that Figure 4 could be improved by visualising only the model surface current velocity vectors co-located with the ADCP observations while retaining the background colour contours representing the surface currents of the entire model

domain. This would allow for more of an 'apples-to-apples' comparison since the authors have already selected data for the observational period between 4th and 22nd January 2024.

> This is also a very good suggestion! We have changed Figure 4 accordingly.

Finally, I believe greater attention should be paid to validating the SCARIBOS model's ability to represent ocean properties, given that communicating the model's overall fidelity will assist future researchers considering using the simulation in their own work. In particular, I would suggest validating the sea surface temperature and sea surface salinity fields outputs by the SCARIBOS model against relevant observations (e.g., OISSTv2 – see Huang et al., 2021 - and the Multi Observation Global Ocean Sea Surface Salinity product – see Droghei et al., 2016).

> We followed the reviewer's suggestions and compared SST and SSS from SCARIBOS with multi-observational products (SST: Guinehut et al., 2012; SSS: Droghei et al., 2016). We incorporated this comparison into the text (lines 177-181 introducing the datasets, and lines 211-228 describing the comparison), making substantial adjustments to Section 2.3, and included the corresponding figures as supplementary material (Figs. S3 and S5).
>
> The agreement in SST is high, while some differences were observed in the SSS fields. To further investigate these discrepancies, we compared SCARIBOS with the GLORYS12V1 model, which provides its initial and boundary conditions. The two models show strong similarities, suggesting that the differences between SCARIBOS and the observational product likely stem from the model dataset used for initialization and boundary forcing. We included comparisons of SST, SSS and surface currents with GLORYS12V1 model in the supplementary material too (Figs. S2, S4 and S6).

Moreover, it also occurred to me that the GO-SHIP / CLIVAR Repeat Section A22 (last completed in 2021) intersects the SCARIBOS model domain, and thus may provide an insightful meridional-cross section with which to compare the model's temperature and salinity field as part of an improved Figure 7 (which does not currently include observations).

> We appreciate this suggestion but have decided not to validate the meridional cross-section at this stage. The A22 section intersects our western boundary, making it less representative of the meridional cross-section studied in Figure 7. Additionally, we plan to include a comprehensive 3D validation of our model in our next manuscript, where we will specifically focus on the 3D ocean properties and circulation in the SCARIBOS simulation.

I would like to emphasise that the suggestions above are intended to give the scientific community even greater confidence in the SCARIBOS model, and hence encourage the wider use of this simulation beyond this particular study (which itself will serve as the documentation and validation of the model going forward).

**Minor Comments**

**Line 298-299**: When does particle seeding conclude in Scenario 3? I could not determine from the current text whether particles are still being released during the final month of the simulation (i.e., February 2024) and, if so, how are these dealt with in the Lagrangian analysis? For example, do you take into consideration the much longer advection time for those particles released in 2020 compared with those released in 2024?

We thank the reviewer for pointing out the potential confusion. To clarify: as stated at the beginning of the paragraph (line 321), particles are seeded from the first until the last day of each month and are tracked until they leave the domain or until the end of the SCARIBOS simulation period. While the final month (February 2024) has less simulation time left, in practice, most particles exit the domain within 30 days, with only about 1% remaining at most. No special treatment was applied to the last month of February 2024. We have added the statement: *"Most particles exit the domain within 30 days of release."* to line 328.

**Figure 8**: Suggest using two separate colorbars for the upper and lower plots given that the colorbar is saturated in the lower plot, but not in the upper plot. This makes the upper plot appear washed out and does not highlight any prominent pathways.

We thank the reviewer for this suggestion. We changed Figure 8 by adding another colorbar.

**Lines 391-394:** Could you quantify this description of strong flushing events somehow? For example, is it the case that particles released in DJFM are more likely (higher proportion) to leave the model domain via the northern and western boundaries compared with those released throughout the rest of the year?

We thank the reviewer for this suggestion. While our study does not aim at quantifying the portion of particles leaving a certain domain boundary, the reviewer is correct that during the strong flushing events more particles are leaving the domain via the northern and western boundaries. We added a sentence to the paragraph in lines 407-408:

> *Conversely, during strong flushing events characterised by strong northwest-directed currents, particles are pushed further away from the southern coastline, resulting in a greater distance from the land for these particles. During such*

*events, particles are also more likely to leave the domain via the western and northern boundaries.*

**Figure 9:** What is special about 2021 for the connectivity of Source regions 6, 7 & 8? This interesting anomalous lack of connectivity feels worthy of comment; what is this related to in the surface circulation?

We thank the reviewer very much for pointing out this valuable observation. We added this observation to the text in lines 435-439 as:

*A notable period from December 2020 to August 2022 stands out, as connectivity from source zones 6, 7, and 8 is very limited, except to their immediate neighbors. This coincides with a prolonged phase of predominantly northwest-directed surface flow (Fig. 6), which remains strong for most of this period. Interestingly, even during months with weaker flow in this period (e.g., May, June 2021), connectivity from these sources remains minimal.*

**Section 3.4:** One intriguing question that came to mind re-reading this Section was: what conditions give rise to the anomalously high and low coastal connectivity values shown between Bonaire and the Venezuelan Islands and Curaçao. We'd certainly care strongly about instances of the former, especially in the context of marine pollution and its wider impacts. Are these episodes of enhanced coastal connectivity predictable?

That is indeed an intriguing question. However, based on the analysis presented in this manuscript, no clear connection can be drawn. For example, the absence of connectivity between Bonaire and Curaçao in July and August 2021, despite predominantly strong northwest-directed currents during these months, suggests that the system is more complex than a straightforward link between regional surface flow direction and observed connectivity patterns. Since our focus has been on surface flow around Curaçao, it is possible that during this period, the current was directed more NNW, causing particles to pass just north of the island. A month later, however, connectivity increased to nearly 50%. Without additional analysis, we cannot fully explain these extreme variations, and we believe that predicting such episodes would require further investigation using additional metrics and an overview of the full domain surface flow.